# Deep Learning for Subspace Regression

**Vladimir Fanaskov**
AXXX, Applied AI Institute
fanaskov.vladimir@gmail.com

**Vladislav Trifonov**
AI4Science, AIC

**Alexander Rudikov**
AXXX, INM

**Ekaterina Muravleva**
AI4Science, AIC

**Ivan Oseledets**
AXXX, INM

## Abstract

It is often possible to perform reduced order modelling by specifying linear subspace which accurately captures the dynamics of the system. This approach becomes especially appealing when linear subspace explicitly depends on parameters of the problem. A practical way to apply such a scheme is to compute subspaces for a selected set of parameters in the computationally demanding offline stage and in the online stage approximate subspace for unknown parameters by interpolation. For realistic problems the space of parameters is high dimensional, which renders classical interpolation strategies infeasible or unreliable. We propose to relax the interpolation problem to regression, introduce several loss functions suitable for subspace data, and use a neural network as an approximation to high-dimensional target function. To further simplify a learning problem we introduce redundancy: in place of predicting subspace of a given dimension we predict larger subspace. We show theoretically that this strategy decreases the complexity of the mapping for elliptic eigenproblems with constant coefficients and makes the mapping smoother for general smooth function on the Grassmann manifold. Empirical results also show that accuracy significantly improves when larger-than-required subspaces are predicted. With the set of numerical illustrations we demonstrate that subspace regression can be useful for a range of tasks including parametric eigenproblems, deflation techniques, relaxation methods, optimal control and solution of parametric partial differential equations.

## 1 Introduction

The goal of reduced order modelling (ROM) is to identify uninformative degrees of freedom and discard them (Bai et al., 2005). The result is a simplified system that is easier to analyse and simulate. This program is computationally demanding and only justified in the setting when many related problems are repeatedly solved and it is possible to use information from encountered problems to build a reduced model for the problems to come. Typical examples are parametric models for partial and ordinary differential equations (PDEs and ODEs), and usual applications are optimisation, sensitivity analysis, uncertainty quantification and control.

As an illustration consider proper orthogonal decomposition (POD) for time-dependent PDEs (Volkwein, 2013), (Hesthaven et al., 2022). To apply POD, one computes solutions for a representative set of parameters and builds a reduced basis for spatial variables by the best low-rank approximation. When new parameters arrive, a computed basis is used to discretise PDE that is solved at reduced cost. In global POD this basis is the same for all incoming parameters and in local POD basis explicitly depends on new parameters. As one may expect, local POD is more expressive than global POD, but can be more challenging to arrange.

POD is an example of the general class of techniques where linear subspace parametrises useful degrees of freedom. While nonlinear ROM techniques exist, linear methods are better understood theoretically, easier to apply in practice, and provide sufficiently well approximation, especially when local versions are available (Franco et al., 2024). In this setting the main challenge is to construct reliable approximation to the function that maps new parameters to linear subspaces.

We analyse this problem under the following assumptions: (i) the set of parameters of interest is specified in form of probability distribution, (ii) it is known how to compute good or optimal linear subspace for each parameter, (iii) the numerically stable method to construct reduced problem from basis is available. In short, we consider regression on grassmannian. We approach the regression problem by specifying loss function and using neural networks as parametric models to accommodate high-dimensional parameter spaces pervasive in practical problems.

More specifically, our main contributions are:

1. Mathematical formulation of subspace regression problem and examples of applications, including eigenspace approximation, local POD, learning basis for deflation and two-grid method, approximating balanced-truncation basis for optimal control problems.

2. Several loss functions, suitable for neural network training, including the stochastic one that scales well with the increase of subspace size.

3. Embedding technique: a strategy to learn a larger subspace containing the target one. Empirically, this technique significantly improves accuracy for subspace learning.

4. Two theoretical justification of embedding technique: derivative of smooth function on Grassmann manifold can be reduced by embedding; complexity of the subspace regression problem for elliptic eigenproblem with constant coefficients.

5. Empirical evaluation of proposed techniques on a diverse set of problems including comparisons with neural surrogates, kernel methods and classical interpolation in normal coordinates.

## 2 SUBSPACE REGRESSION

In this section we formulate precisely what we mean by subspace regression and describe several applications.

### 2.1 DEFINITION OF SUBSPACE REGRESSION PROBLEM

In linear space $\mathbb{R}^n$ we define $k$-dimensional subspace $\mathcal{S}(W) = \left\{ W\alpha, \alpha \in \mathbb{R}^k \right\}$ by specifying tall full rank matrix $W \in \mathbb{R}^{n \times k}$. Matrices $W_1$ and $W_2$ represent the same subspace if there is an invertible matrix $G$ such that $W_1 = W_2 G$. The equivalence class of such matrices is denoted by $\lceil W \rceil$. The set of all $k$-dimensional subspace of $n$-dimensional space $\mathrm{Gr}(k, n)$ is known as Grassmann manifold or grassmanian (Ciaramella et al., 2025), (Bendokat et al., 2024).

Let $V : \mathbb{R}^p \to \mathrm{Gr}(k, n)$ be a function that maps the space of parameters $r \in \mathbb{R}^p$ to the subset of grassmanian represented as the set of tall full rank matrices $V(r)$. We assume that parameters $r$ are sampled from distribution $r \sim p_r$ and that dataset $\mathcal{D} = \{(r_1, V_1), \ldots, (r_m, V_m)\}$ of $m$ i.i.d. samples is available. For a given parametric model $Y_\theta : \mathbb{R}^p \to \mathrm{Gr}(r, n), r \geq k$ we want to identify parameters $\theta^\star$ such that $Y_{\theta^\star}(r)$ approximates $V(r)$[1]. We formulate this task as optimisation problem

$$\theta^\star = \arg\min_\theta \left( \mathbb{E}_{r \sim p_r} \left[ L\left( Y_\theta(r), V(r) \right) \right] \right) \simeq \arg\min_\theta \left( \frac{1}{m} \sum_{i=1}^m L\left( Y_\theta(r_i), V_i \right) \right). \tag{1}$$

Loss function for subspace regression problem is assumed to have two properties:

$$\begin{aligned} &L(A, B) = L(\widetilde{A}, \widetilde{B}) \text{ for arbitrary } \widetilde{A} \in \lceil A \rceil, \ \widetilde{B} \in \lceil B \rceil; \\ &L(A, B) > 0 \text{ and } L(A, B) = 0 \text{ iff } \mathcal{S}(B) \subset \mathcal{S}(A). \end{aligned} \tag{2}$$

In Section 3 we provide explicit expression for loss functions with these properties.

Aside from unusual invariance requirement (2), optimisation problem (1) is a standard machine learning formulation of regression problems which can be solved with stochastic optimisation for arbitrary model $Y_\theta(r)$ that admits efficient evaluation of gradients.

---

[1]Note, that we allow $Y_\theta$ to have more columns than target $V$. In this context approximation is understood in terms of subspace inclusion $\mathcal{S}(V) \subset \mathcal{S}(W_\theta)$. As we explain later, redundancy introduced this way can significantly improve accuracy.

## 2.2 EXAMPLES OF SUBSPACE REGRESSION PROBLEM

**Approximate eigenspaces.** Consider eigenproblem for Schrödinger equation

$$-\Delta\psi(x) + U(x)\psi(x) = E\psi(x), \ \|\psi\|_2 < \infty, \tag{3}$$

where $U(x)$ is potential energy and $E$ is energy of the system. One way to find eigenpairs is to approximate $\psi(x)$ by a finite series $\psi(x) = \sum_{i=1}^{K} \alpha_i\phi_i(x)$ and enforce Petrov-Galerkin condition that residual is orthogonal to all $\phi_i(x)$. Continuous problem (3) reduces to eigenproblem for Hermitian matrix and can be solved in $O\left(K^3\right)$ operations (Trefethen & Bau, 2022). For eigenproblems we are typically interested only in extremal eigenspaces corresponding to either smallest or largest eigenvalues. In this case it is desirable to select a small number of basis functions $\phi_i(x)$ that approximate sufficiently well the subspace of interest. When eigenproblem (3) is solved repeatedly for many potential functions $U(x)$ this lead us to subspace regression problem (1) used to approximate the mapping $U(x) \to \left\{f(x) : f(x) = \sum_{i=1}^{K} \alpha_i\phi_i(x), \alpha_i \in \mathbb{C}\right\}$.[2] That is, we wish to predict subspace spanned by first $K$ eigevectors. When this mapping is learned from a set of examples, eigenproblems for unobserved potentials $U$ can be solved efficiently, since low-dimensional candidate subspace for eigenfunctions is available.

**Intrusive POD for time-dependent PDEs.** As an example of time-dependent PDE we use Burgers equation

$$\frac{\partial u(x,t)}{\partial t} + u(x,t)\frac{\partial u(x,t)}{\partial x} = \frac{\partial}{\partial x}\left(\nu(x)\frac{\partial u(x,t)}{\partial x}\right), \ u(0,t) = u(1,t) = 0, \ u(x,0) = u_0(x). \tag{4}$$

One starts with spatial discretisation which reduces equation (4) to the set of ODEs and define inner product $\langle\cdot,\cdot\rangle_W$ for discretised $u(t)$ that approximates $L_2$ inner product. For this set of ODEs the reduced degrees of freedom $\psi_i$ are defined as solution to optimisation problems $\min \int_0^T dt \, \|u(t) - \langle\psi_i, u(t)\rangle_W \psi_i\|_W^2$ subject to $\langle\psi_i, \psi_j\rangle = 0, j < i, \ \langle\psi_i, \psi_i\rangle = 1$. When discretised, this scheme lead to optimal basis computed with SVD from snapshot matrix (Volkwein, 2013). This basis can only be computed when equation (4) is integrated, so POD is justified only in situation when many related problems are solved. We apply subspace regression with POD to learn the function that maps PDE data to the subspace formed by reduced basis $\{\psi_1, \ldots, \psi_k\}$ for some small $k$. Notably, this allows us to apply local POD to high-dimensional parametric problems.

**Coarse grid correction for iterative methods.** Consider stationary diffusion equation with Dirichlet boundary conditions

$$-\text{div } k(x) \text{ grad } \phi(x) = f(x), \ x \in \Gamma, \ \phi(x)|_{\partial\Gamma} = 0. \tag{5}$$

When equation (5) is discretised with finite difference or finite element method, it reduces to linear problem $A\phi = f$, where $A$ is large sparse matrix and $\phi, f$ are discretised solution and right-hand side of (5). To exploit sparsity of $A$ one can solve linear equation with relaxation method. General relaxation method split matrix additively $A = D + C$, where $D$ is regular with known inverse (Saad, 2003). Given the split, if iteration scheme $x^{n+1} = x^n + D^{-1}(b - Ax^n)$ is convergent, steady state is exact solution $x = A^{-1}b$. Convergence is linear and its rate is defined by spectral radius of error propagation matrix $I - D^{-1}A$. To improve convergence rate, one can augment iterative method with coarse-grid correction (Trottenberg et al., 2001). This techniques allows one to remove influence of leading subspace formed by colums of matrix $V$ of $I - D^{-1}A$ by solving small reduced linear system for error equation $V^\top AVe = r$, where $e$ and $r$ are error and residual in the subspace $\mathcal{S}(V)$. Naturally, subspace regression (1) for this problem approximates the mapping $A \to \mathcal{S}(V)$ or $k(x) \to \mathcal{S}(V)$ for linear systems resulting from equation (5).

**Deflation for conjugate gradient.** Krylov subspace methods provide a more systematic way to solve large sparse linear systems (Saad, 2003). For linear system with symmetric positive definite matrix $A$ resulting from discretisation of equation (5), the method of choice is conjugate gradient (CG) (Hestenes et al., 1952). Similar to other Krylov methods, on step $r$, CG identify optimal solution within Krylov subspace $\mathcal{K}_r = \text{span}\left\{b, Ab, \ldots, A^{r-1}b\right\}$, where span refers to the subspace formed by linear combinations of vectors in the set. Since powers of $A$ are involved, the most readily available vectors are from the subspaces with large eigenvalues (Saad, 2011). To improve

---

[2]Suitable discretisation of parametrisation of both $\phi$ and $U$ is assumed.

convergence of method it is reasonably to include eigenspaces $V$ with small eigenvalues to the approximation space $\mathcal{K}_r$. The resulting method is deflated CG and the approximation space is $\mathcal{K}_r \cup \mathcal{S}(V)$ (Saad et al., 2000). In analogy with previous example, the subspace regression problem considered approximates $A \to \mathcal{S}(V)$ or $k(x) \to \mathcal{S}(V)$, but this time $V$ spans eigenspaces with small eigenvalues of matrix $A$.

**Balanced-truncation for linear-quadratic control.** Suppose we want to solve the following linear quadratic control problem

$$\dot{y}(t) = Ay(t) + Bu(t),\ z(t) = Cy(t),$$

$$J = \int dt \left( (z(t))^\top Qz(t) + (u(t))^\top Ru(t) \right) + (z(T))^\top Mz(T), \tag{6}$$

where $y(t)$ is state variable, $u(t)$ is control, $z(t)$ is observable, $A, B, C, Q, R, M$ are matrices of appropriate sizes, $R$ is symmetric positive definite, $Q$ and $M$ are symmetric positive semidefinite. The goal is to find a control signal $u(t), t \in [0, T]$ that minimises cost function $J$.

In the situation when number of state variables $y(t)$ is large, one may want to apply ROM to compute optimal control at a reduced cost. An established way to do that is balanced truncation (Moore, 2003). Roughly speaking, balanced truncation compute a special coordinate system $y(t) = \mathcal{T}\widetilde{y}(t)$ that discounts variables that are both unobservable and uncontrollable, so only a few first columns of matrix $\mathcal{T}$ can be used to accurately model (6). This is done by finding coordinate system that simultaneously diagonalises observability gramian $G_o$ and controlability gramian $G_c$ defined as solutions of Lyapunov equations $A^\top W_o + W_o A + C^\top C = 0$, $AW_c + W_c A^\top + BB^\top = 0$ (Moore, 2003), (Volkwein, 2013). In this case the goal of subspace regression (1) is to approximate the mapping $A, B, C \to \mathcal{S}\left(\overline{\mathcal{T}}\right)$, where $\overline{\mathcal{T}}$ is tall matrix assembled from first few columns of $\mathcal{T}$.

## 3 THEORETICAL RESULTS

We proceed by characterising loss functions, introducing the subspace embedding technique and providing its theoretical justification.

### 3.1 LOSS FUNCTIONS

Requirements (2) that allow loss function to work with $\mathrm{Gr}(k, n)$ data enforce right $\mathrm{GL}(k)$ invariance. As a consequence all loss functions introduced below are all based on orthogonal projectors.

**Theorem 1.** *Let $A \in \mathbb{R}^{n \times k}$, $B \in \mathbb{R}^{n \times p}, p \leq k$ be tall full rank matrices.*

1. *Loss function $L_1(A, B) = p - \left\| Q_B^\top Q_A \right\|_F^2$ satisfies requirements (2), where $A = Q_A R_A, B = Q_B R_B$ are reduced QR decompositions of $A$ and $B$, $\|\cdot\|_F$ is Frobenius norm[3].*

2. *Let $z \in \mathbb{R}^k$ be a random variable with zero mean and identity covariance matrix. Loss functions $L_2(A, B; z) = \min_u \|Au - Q_B z\|_2^2$ does not depend on the choice of $A$ from $\lceil A \rceil$, where $B = Q_B R_B$ is QR decomposition.*

3. *On average $L_2$ equals $L_1$, i.e., $\mathbb{E}_z \left[ L_2(A, B; z) \right] = L_1(A, B)$.*

*Proof.* Appendix A. ☐

Loss $L_1$ is essentially the same as the difference of orthogonal projectors. Note, that $L_1(A, B) \geq 0$ with equality reached if and only if matrices $A$ and $B$ share the same columns space, since in this case $\left\| Q_B^\top Q_A \right\|_F^2 = p$. Loss $L_2$ introduces two modifications: (i) projector in Riemannian distance is replaced with error of least squares problem; (ii) to remove second projector, stochastic Hutchinson trace estimation is used. Reformulation with least square problem allows one to use normal equation, and various tools from randomised numerical linear algebra, e.g., randomised

---

[3]Reduced QR decomposition of tall full rank matrix $A \in \mathbb{R}^{n \times k}$ is a factorisation $A = Q_A R_A$, where $Q_A \in \mathbb{R}^{n \times k}$ has orthonormal columns, $R_A \in \mathbb{R}^{k \times k}$ is upper triangular with nonzero elements on the diagonal.

preconditioned Cholesky-QR (Garrison & Ipsen, 2024), blendenpik solver (Avron et al., 2010), and sketching (Woodruff et al., 2014). We will see in Section 4, that loss function $L_2(A, B)$ based on normal equation scales better than $L_1(A, B)$ with the increase of subspace size.

## 3.2 SUBSPACE EMBEDDING

In the definition of subspace regression problem (1) we allow to approximate function $\mathbb{R}^p \to Gr(k, n)$ by function $\mathbb{R}^p \to Gr(r, n)$ where $r \geq k$. We call this strategy subspace embedding. It is justified because of two unique properties of regression and interpolation on grassmanian: (i) inclusion of vector subspaces is well-defined; (ii) subspace, predicted by regression model or interpolated by standard techniques, is used to construct a reduced model. From the latter property one may expect similar or improved accuracy when the predicted subspace from $Gr(r, n)$ contains target subspace from $Gr(k, n)$.

We will show empirically in Section 4 that subspace embedding significantly improves accuracy and generalisation gap. Here we argue that prediction of larger-than-needed subspaces align well with inductive bias of neural networks known as f-principle or spectral bias (Xu et al., 2019). F-principle is an observation that neural networks tend to learn smoothed versions of the target functions. As we show below, embedding techniques may improve smoothness of learned function.

**Theorem 2.** *Let $V : \mathbb{R} \to Gr(k, n)$ be continuously differentiable on $t \in [0, T]$, $V(t)^\top V(t) = I_k$. It is always possible to construct piecewise continuous function $W : \mathbb{R} \to Gr(r, n)$, $r > k$, $W(t)^\top W(t) = I_r$ such that $\frac{1}{2} \left\| W(t)W(t)^\top - V(t)V(t)^\top \right\|_F^2 - \frac{r-k}{2}$ is arbitrary small and $\left\| \dot{W}(t) \right\|_F^2 \leq \left\| \dot{V}(t) \right\|_F^2$, where inequality is strict for all points where $\left\| \dot{V}(t) \right\|_F^2 \neq 0$.*

*Proof.* Appendix B; See Appendix C for subspace embedding example. □

Theorem 2 implies that, by increasing the subspace size, one can always approximate continuously differentiable functions arbitrarily well and simultaneously reduce its derivative. F-principle suggests that the latter property makes learning easier for neural networks.

## 3.3 COMPLEXITY OF PARAMETRIC EIGENPROBLEM

To illustrate difficulties one may encounter and to further justify embedding technique we consider complexity of subspace regression problem for parametric elliptic eigenproblem with constant coefficient

$$-\sum_{i=1}^{D} a_i \frac{\partial^2 \phi_{i_1,\ldots,i_D}(x_1,\ldots,x_D)}{\partial x_i^2} = \lambda_{i_1,\ldots,i_D} \phi_{i_1,\ldots,i_D}(x_1,\ldots,x_D), \tag{7}$$

where $a_i > 0$, $x_i \in [0, 1]$ and Dirichlet boundary conditions are assumed.

For problem (7) general eigenfunction is $\phi_{i_1,\ldots,i_D}(x_1,\ldots,x_D) = \prod_{j=1}^{D} \sin(\pi i_j x_j)$ and the set of eigenfunctions does not depend on coefficients $a_i$. Observe that $\lambda_{i_1,\ldots,i_D} = \sum_{j=1}^{D} a_j (\pi i_j)^2$, so coefficients $a_i$ define the order of eigenvectors. Below we formally characterise mapping from coefficients to $k$-th eigenvector and eigenspace.

**Theorem 3.** *Suppose eigenvectors of (7) are ordered according to the increase of eigenvalues. Let $\phi_k$ be an eigenvector on position $k$, let $V_k$ be an eigenspace spanned by vectors on positions up to $k$. Consider mappings $F_k : a_1,\ldots,a_D \to \phi_k$ and $G_k : a_1,\ldots,a_D \to V_k$.*

1. *$F_k, G_k$ are piecewise constant functions that map real numbers to elements of sets $S_{F_k}, S_{G_k}$. Sets $S_{F_k}, S_{G_k}$ are finite with $\#_{F_k}(k, D), \#_{G_k}(k, D)$ distinct elements.*

2. *Let $W_l, l > 1$ be a subspace obtained by union of $V_l$ for distinct $a_1,\ldots,a_D$. Number of vectors in $W_l$ is $\#_{F_k}(l, D) + 1$.*

3. *$\#_{F_k}(k, D) \sim \frac{1}{(D-1)!} k (\log k)^{D-1}$ for fixed $D$ and large $k$.*

4. *$\#_{F_k}(k, D) \leq k D^{\log_2 k}$ for fixed $k$ and large $D$.*

5. $\#_{G_k}(k, D) \geq \frac{1}{(D-1)!} k^{D-1}$ *for fixed D and large k.*

6. $\#_{G_k}(k, D) \geq \frac{1}{(k-1)!} D^{k-1}$ *for fixed k and large D.*

*Where $\sim$ is asymptotic expansion and $\geq, \leq$ are lower and upper bound on leading asymptotic.*

*Proof.* Appendix D; See Appendix E for examples. □

Theorem 3 suggests that for problem (7) mappings from coefficient to $k$-th eigenvector $\phi_k$ or subspace $V_k$ are piecewise constant functions with rapidly growing number of constant regions when either $k$ or $D$ increases. The complexity (the number of regions) of the subspace prediction problem exceeds the complexity of $k$-th eigenvector prediction. However, results also suggest that the number of unique eigenvectors within $V_k$ grows at the same rate as the number of eigenvectors on position $k$. This means, the large number of distinct regions in $G$ comes from a large number of possible combinations of an asymptotically small number of vectors. Given that, the complexity of mapping from coefficients to subspace decreases, if one predicts subspace of larger dimension $\widetilde{V}_k \supseteq V_k$. For example, if one is willing to predict subspace of dimension $\#_F(k, D)$ in place of $V_k$ of dimension $k$, the mapping $a_1, \ldots, a_D \to \widetilde{V}_k \supseteq V_k$ may be chosen to have constant value.

## 4    NUMERICAL EXPERIMENTS

We present several numerical experiments to corroborate our theoretical findings. The discussion of control problems appears in Appendix I. All numerical results are reported as single-run metrics without explicit error bars. To study sensitivity to train-test split we perform several dedicated experiments. Results suggest that variance is low and does not affect main conclusions. The details are available in Appendix J. For most problems we report relative error measured in percents. For predicted quantity $\widetilde{v}$ and ground truth value $v$ it reads $100\% \times \|v - \widetilde{v}\|_2 / \|v\|_2$. For eigenvalue problems the numerator is $\mathbb{Z}_2$ adjusted as explained below. For iterative methods we report convergence plots for relative error, without additional factor $100\%$.

### 4.1    EIGENSPACE PREDICTION

We considered several eigenvalue problems: (i) $D = 1$ eigenproblem with Schrödinger operator (3) with parametric family of expanded Morse oscillator (Le Roy et al., 2006), (ii) $D = 2$ Schrödinger operator (3) with parametric family of two expanded Morse oscillators (Carpenter et al., 2018), (iii) $D = 2$ two datasets, $k_1 = k_2$ and $k_1 \neq k_2$, for elliptic eigenproblem (5) (left-hand side of the equation) with contrast coefficient sampled from gaussian random field, (iv) $D = 3$ dataset for elliptic eigenproblem with diffusion coefficient $k_1 = k_2$. In all experiments we used FFNO architecture (Tran et al., 2021) , a modification of FNO (Li et al., 2020), and performed extensive hyperparameter grid search. Details on dataset generation and training protocol are available in Appendix F. To contextualise subspace regression we provide results for two baselines.

**Regression with $\mathbb{Z}_2$ adjusted $l_2$ loss.** Eigenvectors are defined up to a sign, so in place of subspace losses specified in Theorem 1 one can try to directly predict eigenvectors with $\mathbb{Z}_2$ adjusted $l_2$ loss $l_{\mathbb{Z}_2}(v, u) = \min_{\pm} \|v \pm u\|_2$.

**Interpolation in Riemannian normal coordinate system.** A standard technique of manifold interpolation applied to grassmannian (Amsallem, 2010), (Ciaramella et al., 2025), (Zimmermann, 2019). For a given query, $k$ closest points are selected from the training set. One point supplies common tangent space, i.e., it is used to compute logarithms for the remaining points. Since logarithms lay in the same tangent space they can be interpolated with any techniques desirable (we use RKHS (Bishop & Nasrabadi, 2006)). After interpolation of logarithms, the exponential map is computed.

Additional results are available in Appendix F. Here we highlight several interesting trends.

**Subspace losses are unsuccessful without subspace embedding technique.** Figure 1b contains results of learning subspace spanned by first 10 eigenvectors for $D = 2$ on grid $100 \times 100$ elliptic eigenproblem with $L_2(A, B; z)$ (loss $L_1(A, B)$ leads to the same accuracy). Neural network predicts

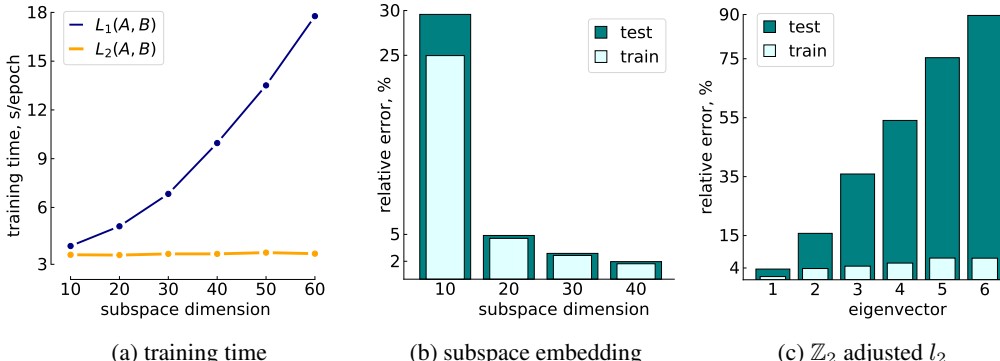

(a) training time        (b) subspace embedding        (c) $\mathbb{Z}_2$ adjusted $l_2$

Figure 1: Selected results for eigenspace prediction: (a) Comparison of training time for losses $L_1(A,B)$, $L_2(A,B;z)$ from Theorem 1. On $D = 2$ grid $N_x = N_y = 100$ we observe $L_2(A,B;z)$ scales better with dimension size; (b) Illustration of subspace embedding technique from Section 3.2 for $D = 2$ elliptic eigenproblem, prediction of first 10 eigenvectors. Prediction of larger subspace manifestly improves accuracy and reduces generalisation gap; (c) Relative error for individual eigenvector predictions for the same problem as in (b) but trained with $\mathbb{Z}_2$-adjusted $l_2$ loss. Similarly to results of Theorem 3 we observe a steep increase of problem complexity with eigenvector number. See Section 4.1 for details.

subspace of sizes 10, 20, 30, 40 according to subspace embedding strategy Section 3.2. Results demonstrate subspace embedding is efficient in decreasing test error from 30% for subspace of dimension 10, to test error 2% for subspace of dimension 40 (0.4% of the total number of degrees of freedom). It is less clear from Figure 1b, but the generalisation gap also systematically improves, suggesting that complexity of the problem decreases. Similar conclusions are valid for $D = 2$ QM and $D = 3$ eigenproblems.

**Classical interpolation is not competitive.** In Table 1 we gather results (best for each method) for one $D = 1$ and two $D = 2$ QM problems. Classical interpolation is reasonably accurate only on the simplest problem in $D = 1$, but $L_{\mathbb{Z}_2}$ loss still results in better accuracy. The reason is likely classical interpolation struggles in high-dimensional subspaces because observations are too sparse to naively approximate tangent space in the region of interest by finding nearest neighbours. On QM datasets accuracy of subspace regression is consistently better than for other approaches.

Table 1: Relative errors for QM problems.

| dataset | interp. | $L_{\mathbb{Z}_2}$ | $L_1(A,B)$ |
|---|---|---|---|
| $D = 1$ | 4.69% | 2.33% | 0.09% |
| $D = 2, a$ | 31.9% | 19.52% | 0.65% |
| $D = 2, b$ | 92.64% | 48.56% | 15.58% |

**Loss without $QR$ scales much better for larger subspace sizes.** In Figure 1a we demonstrate wall clock training time for $L_1(A,B)$ and $L_2(A,B;z)$ (least squares problem is solved with normal equation) per epoch on the same hardware for the same FFNO architectures. For small subspace sizes the training time the methods are roughly on par, but with the increase of subspace size QR starts to drastically slow down training with $L_1$ loss.

**Training with loss $L_{\mathbb{Z}_2}$ is reasonable only for several first eigenvectors.** In Figure 1c we present results for learning individual eigenvectors (a separate network is trained for each eigenvector) for $D = 2$ elliptic eigenproblem.Train error is reasonably small for all eigenvectors, which imply neural networks can successfully approximate them. Rapid growth of the test error with eigenvector number indicates the increase of problem complexity in agreement with Theorem 3.

**Neural networks trained with subspace embedding technique learn smoother maps.** Theorem 2 suggests it is possible to decrease derivative by embedding of geodesics into a larger space. This provides only a circumstantial evidence that the same may happen when neural networks are trained with subspace embedding technique. In Appendix F we gather empirical results that support such conclusion. The results are based on several "smoothness indicators": the error of linear model,

Frobenius norm of derivative, and mean cosine of angles between subspaces at nearby points. We refer interested readers to Appendix F.5.

**Subspace regression can speed-up classical iterative eigensolvers.** As an example of hybrid approach we consider combination of subspace regression and LOBPCG (Knyazev, 2001). LOBPCG is a classical iterative matrix-free eigensolver that can approximate extremal eigenspaces. To apply it in combination wit hsubspace regression we use trained neural network to predict subspace, and initialise LOBPCG with solution of reduced eigenproblem. Note, that the cost of such initialisation is negligible small comparing to the full cost of LOBPCG iterations. We observe 2 to 3 times faster converges and 2 orders lower relative error on average when subspace regression is used for initialisation. More details are available in Appendix F.6.

**Loss $L_2(A, B; z)$ may become unstable.** From results summarised in Figure 1a one can assume that $L_2(A, B; z)$ is always preferable. Results for $D = 3$ problem (elliptic eigenproblem, grid $30 \times 30 \times 30$, prediction of first 3 eigenvectors) summarised in Table 2 reveal a more nuanced picture. Loss $L_2(A, B; z)$ clearly performs worse than $L_1(A, B)$ and even fails for subspace size 24. The reason for that is numerical instability of solvers based on the normal

Table 2: Accuracy for $D = 3$ elliptic eigenproblem.

| $N_{\text{sub}}$ | $L_1(A, B)$ | $L_2(A, B; z)$ | $L_2^{\text{stab}}(A, B; z)$ |
|---|---|---|---|
| 6 | 24.77% | 31.46% | 28.28% |
| 12 | 13.69% | 17.12% | 15.88% |
| 24 | 9.71% | – | 9.49% |
| 48 | 7.54% | 16.3% | 7.4% |

equation. To stabilize $L_2(A, B; z)$ we apply Cholesky-QR2 (Yamamoto et al., 2015). The results for stabilised loss shows that accuracy becomes comparable to $L_1(A, B)$ and even slightly better for larger subspace dimensions.

## 4.2 Parametric PDE problems

We considered two PDEs: (i) $D = 1 + 1$ viscous Burgers equation, related to benchmark from (Li et al., 2020); (ii) $D = 2$ elliptic problems (5). Our main operator is FFNO and the solutions strategy we use is classical intrusive POD[4]. For datasets description and training details see Appendix G. We compare subspace regression with several methods.

**Regression with FFNO.** We apply FFNO, an extension of Fourier Neural Operator, to parametric PDEs in a standard way similar to (Tran et al., 2021).

**Regression with DeepONet.** Classical architecture based on the universal approximation theorem of operators (Lu et al., 2019). DeepONet can be understood as end-to-end training of non-intrusive POD with basis functions parametrised by implicit neural representation or physics-informed neural networks (Sitzmann et al., 2020), (Lagaris et al., 1998), (Raissi et al., 2019).

**Intrusive POD with DeepONet/FFNO basis.** When DeepONet is trained, learned spatial or spatiotemporal basis functions can be used to extract basis (by method directly related to POD) suitable for spectral methods or intrusive POD (Meuris et al., 2021), (Meuris et al., 2023). As suggested in the discussion section of (Meuris et al., 2021), the same can be done with FNO.

**Deep POD.** Projector-based loss is used directly to extract basis from available snapshot matrices or steady-state solutions (Franco et al.). In the referenced publication this approach is combined with PCA-Net described below.

**Kernel methods.** A non-parametric technique where the RKHS method is used for encoder, processor and decoder (Batlle et al., 2024).

**PCA-Net.** A non-intrusive technique with classical POD used as both encoder and decoder, and feedforward network served as processor (Hesthaven & Ubbiali, 2018), (Bhattacharya et al., 2021).

**POD and oracle.** Two POD-based baselines. POD is an intrusive variant of global POD (Volkwein, 2013). Oracle is an intrusive variant of local POD computed with optimal subspace. In problems we consider, error achieved by oracle is the best possible for a given number of basis vectors.

---

[4]Recall, that when coefficients in the reduced basis expansion are predicted by some model we have non-intrusive POD. When basis is used to generate reduced ODE that is later integrated we have intrusive POD.

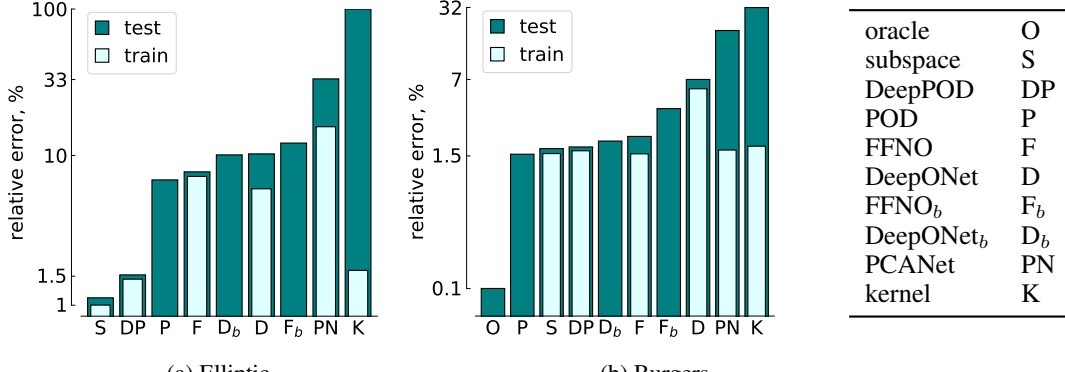

(a) Elliptic.            (b) Burgers.

Figure 2: Relative errors for selected baselines. Label "subspace" refers to subspace regression. For the elliptic problem (a) subspace dimension of ROM methods is bounded by $100$, and for DeepPOD, and subspace regression – by $40$. Oracle is omitted for the elliptic problem because it has perfect accuracy with $10$ basis functions. For Burgers equation subspace dimensions for all methods $\leq 50$. $FFNO_b$ and $DeepONet_b$ refer to an intrusive ROM with bases extracted from FFNO and DeepONet.

Additional results are available in Appendix G. Here we highlight the most important findings.

**Subspace regression leads to competitive accuracy.** In Figure 2a and Figure 2b we observe that subspace regression performs similar or better than DeepPOD. Among other intrusive methods only classical POD leads to comparable accuracy. Bases extracted from FFNO and DeepONet are worse than global POD in all experiments. Pure regression approaches – FFNO, DeepONet, PCANet – appear to be less accurate. PCANet similar to kernel methods shows significant overfitting on our problems, likely resulting from poor inductive bias of the architecture. Note however, that regression approaches are not directly comparable with intrusive techniques, since they do not require a solution of reduced model.

**Representations learned by neural networks are highly non-optimal.** Representation of all neural networks are inefficient if one compares them with the oracle. For example, by construction of an elliptic dataset, a subspace of dimension $10$ is enough for perfect accuracy. Both DeepPOD and subspace regression reach error about $< 1.5\%$ with subspaces of dimension $40$, DeepONet needs to have $> 200$ basis functions to reach comparable accuracy, and FFNO with $64$ basis functions in the last hidden layer lead to $10\%$ relative error. Basis functions built from FFNO and DeepONet are similarly inefficient. The same observations hold for Burger's equation.

### 4.3 ITERATIVE METHODS FOR LINEAR SYSTEMS

We illustrate subspace regression for iterative methods using $D = 2$ elliptic problems (5). Said iterative methods are deflated CG and two-grid correction for the Jacobi method introduced in Section 2 and explained in more detail in Appendix H. Figure 3a and Figure 3b shows average convergence curves on test set and Appendix H contains the rest of relevant data.

**Iterative methods are less sensitive to subspace quality.** On the training stage, neural networks were presented with data only on first $10$ eigenvectors. Despite that, neural networks trained with subspace embedding nearly match the performance of deflated CG with exact eigenspaces of larger size, for coarse-grid corrected Jacobi method convergence speed with learned subspaces is even slightly better. One possible explanation hinted by Theorem 3 is that from distribution of subspaces some information about nearby vectors can be recovered.

**Seemingly minor variations in problem setting can lead to substantial variations in the complexity of the learning problem.** Initially for the Jacobi method we posed a subspace regression problem as an approximation of leading eigenspaces of error propagation matrix $I - D^{-1}A$, where $D$ is diagonal of $A$. Neural networks with and without subspace embedding completely failed to learn. After inspection of the dataset we found that the leading eigenspace contains a complicated mixture of functions with low and high frequencies. Since the learning problem appeared to be completely

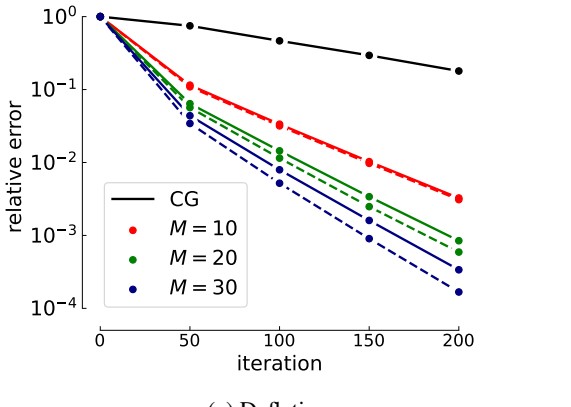 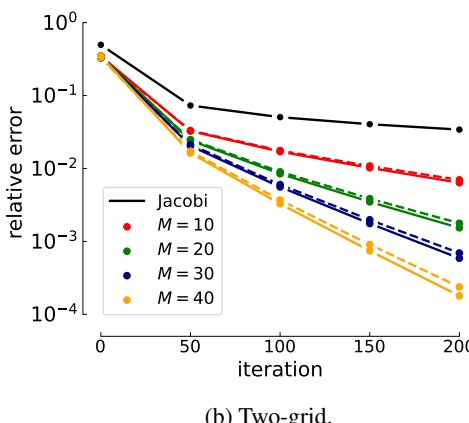

(a) Deflation (b) Two-grid.

Figure 3: Convergence results for iterative methods. Learned methods are marked with solid lines, and dashed lines correspond to iterative methods with optimal deflation and coarse-grid spaces, $M$ refers to subspace size.

hopeless, we reformulated subspace regression using error propagation matrix of damped Jacobi iteration $I - \omega D^{-1} A$ with $\omega = 0.9$. In contrast to the standard Jacobi method, the damped version leads to subspaces formed by low frequency functions. As evident from Figure 3b the resulting mapping is easily learnable. A more detailed report can be found in Appendix H.

## 5 CONCLUSION

Subspace regression – a prediction of subspace from available data – is an interesting problem with a variety of applications including reduced order modelling for partial differential equations, approximation of eigenspaces for eigenproblems, construction of iterative methods for linear problems and optimal control. We formalise subspace regression as a statistical learning problem and introduce several loss functions that are suitable for subspace data. For most of the applications considered we observe that the learning problem is too complicated even when a specialised loss function is used. To simplify learning we propose to approximate a given subspace with a subspace of larger dimension. The resulting technique, called subspace embedding, significantly improves accuracy and generalisation gap. The idea of subspace embedding is that redundancy typically simplifies the learning process and leads to more robust performance. Even though this strategy clearly helps, it introduces a large gap between dimensions of optimal and learned subspaces. The same gap is observed for the classical operator learning problems, when the neural network is trained to approximate solution mapping for parametric PDE. In this case the learned basis can be extracted from the last hidden layer. This neural basis is far from optimal, requiring an excessive number of basis vectors to be used for reaching comparable accuracy. Whether this inefficiency in representation can be resolved, remains an open problem.

## 6 REPRODUCIBILITY STATEMENT

Code used for training, evaluation, and dataset generation is available on `https://github.com/VLSF/subreg`. In the current version detailed description of architectures, hyperparameters, dataset generation and training details appear in Appendix I, Appendix H, Appendix G, Appendix F.

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

## A  PROOF OF THEOREM 1

1. To show that $L_1(A, B)$ does not depend on the chosen representative we observe that

$$L_1(A, B) = p - \left\| Q_B^\top Q_A \right\|_F^2 = \frac{1}{2} \left\| P_B - P_A \right\|_F^2 - \frac{k-p}{2}, \tag{8}$$

where $P_A = A \left( A^\top A \right)^{-1} A^\top, P_B = B \left( B^\top B \right)^{-1} B^\top$ are orthogonal projectors on the columns spaces of $A$ and $B$. When QR decompositions $A = Q_A R_A, B = Q_B R_B$ are available, projectors become $P_A = Q_A Q_A^\top, P_B = Q_B Q_B^\top$ and identity (8) can be verified by algebraic manipulations

$$\frac{1}{2} \left\| P_B - P_A \right\|_F^2 - \frac{k-p}{2} = \frac{1}{2} \text{tr} \left( \left( Q_B Q_B^\top - Q_A Q_A^\top \right) \left( Q_B Q_B^\top - Q_A Q_A^\top \right) \right) - \frac{k-p}{2}$$

$$= \frac{1}{2} \text{tr} \left( Q_B Q_B^\top \right) + \frac{1}{2} \text{tr} \left( Q_A Q_A^\top \right) - \left\| Q_B^\top Q_A \right\|_F^2 - \frac{k-p}{2} = p - \left\| Q_B^\top Q_A \right\|_F^2. \tag{9}$$

From the equivalent form of loss $L_1(A, B)$ given in equation (8) one can immediately conclude that $L_1(A, B)$ does not depend on the representatives $A, B$ chosen from $\lceil A \rceil, \lceil B \rceil$. The reason is projectors are invariant under right GL transformations. For example, $P_A = P_{\widetilde{A}}$, where $\widetilde{A} = AG$ and $G$ is arbitrary non-degenerate matrix $G \in \mathbb{R}^{k \times k}$

$$\widetilde{A} \left( \widetilde{A}^\top \widetilde{A} \right)^{-1} \widetilde{A}^\top = AG \left( G^\top A^\top AG \right)^{-1} G^\top A^\top$$

$$= AGG^{-1} \left( A^\top A \right)^{-1} \left( G^\top \right)^{-1} G^\top A^\top = A \left( A^\top A \right)^{-1} A^\top. \tag{10}$$

Now, when we know that $L_1(A, B)$ does not depend on the chosen representatives, it is easy to show that the minimal value of loss is 0 and it is reached when $\mathcal{S}(B) \subset \mathcal{S}(A)$. To see this, select representatives such that $Q_A = \left( \widetilde{Q}_B \quad \widetilde{Q}_B^\perp \right)$, where $\widetilde{Q}_B$ is block matrix formed from subset of columns of $Q_B$ and columns of $\widetilde{Q}_B^\perp$ are all orthogonal to $Q_B$. This selection is always possible since $\left( I - Q_B Q_B^\top \right) + Q_B Q_B^\top = I$. Representatives selected in this form give

$$L_1(A, B) = p - \left\| Q_B^\top \widetilde{Q}_B \right\|_F^2 = p - q \geq 0, \tag{11}$$

where $\widetilde{Q}_B \in \mathbb{R}^{n \times q}, q \leq p$. The last identity follows by construction: $\widetilde{Q}_B$ is composed from columns of $Q_B$. Loss becomes zero only if $p = q$, or, equivalently, $\mathcal{S}(B) \subset \mathcal{S}(A)$.

2. We first show that

$$L_2(A, B; z) = \min_u \left\| Au - Q_B z \right\|_2^2 = \left\| \left( I - P_A \right) Q_B z \right\|_2^2, \tag{12}$$

where $P_A = A \left( A^\top A \right)^{-1} A^\top$ is orthogonal projector on the columns space of $A$. Using $I = \left( I - P_A \right) + P_A$, and $A \left( I - P_A \right) = \left( I - P_A \right) A = 0$ we obtain

$$\min_u \left\| Au - Q_B z \right\|_2^2 = \min_u \left\| Au - P_A Q_B z - \left( I - P_A \right) Q_B z \right\|_2^2$$

$$= \min_u \left\| Au - P_A Q_B z \right\|_2^2 + \left\| \left( I - P_A \right) Q_B z \right\|_2^2 = \left\| \left( I - P_A \right) Q_B z \right\|_2^2. \tag{13}$$

The last equality holds since $P_A Q_B$ and $A$ share the same columns space. Given that $P_A$ does not depend on representative $A$ from $\lceil A \rceil$, and that $L_2(A, B; z)$ depends on $A$ only via $P_A$, we conclude that the same is true for $L_2(A, B; z)$.

3. From equation (12) we find

$$\mathbb{E}_z \left[ L_2(A, B; z) \right] = \mathbb{E}_z \left[ \left\| \left( I - P_A \right) Q_B z \right\|_2^2 \right] = \mathbb{E}_z \left[ z^\top \left( Q_B^\top \left( I - P_A \right) Q_B \right) z \right]$$

$$= \mathbb{E}_z \left[ \text{tr} \left( \left( Q_B^\top Q_B - Q_B^\top Q_A Q_A^\top Q_B \right) z z^\top \right) \right] = \text{tr} \left( \left( Q_B^\top Q_B - Q_B^\top Q_A Q_A^\top Q_B \right) \mathbb{E}_z \left[ z z^\top \right] \right)$$

$$= p - \left\| Q_B^\top Q_A \right\|_F^2 = L_1(A, B). \tag{14}$$

## B    PROOF OF THEOREM 2

We provide two comments before proceeding with the proof.

In most parts of the text we assumed working with the non-compact Stiefel manifold and in this theorem we have data on the compact Stiefel manifold (see (Amsallem, 2010) for definitions). We specify how one can compute $Q_A$ and $\dot{Q}_A$ having $A$ and $\dot{A}$. One may start from any stable version of Cholesky QR, e.g., (Garrison & Ipsen, 2024), (Yamamoto et al., 2015), and obtain

$$Q_A = AR^{-1}, \tag{15}$$

where $R$ is Cholesky factorization of Gram matrix $A^\top A$, i.e., $A^\top A = R^\top R$ where $R$ is a lower triangular square invertible matrix. To find the derivative of $Q_A$ we need to know the derivative $\frac{d}{dt}R^{-1}$. Derivative $\dot{R}$ can be computed as a solution to Lyapunov equation

$$\dot{R}^\top R + R^\top \dot{R} = \dot{A}^\top A + A^\top \dot{A}, \tag{16}$$

after that $\frac{d}{dt}R^{-1}$ can be found from Jacobi identity $\frac{d}{dt}R^{-1} = -R^{-1}\dot{R}R^{-1}$.

In Theorem 2 we use $\frac{1}{2}\left\|W(t)W(t)^\top - V(t)V(t)^\top\right\|_F^2 - \frac{r-k}{2}$ to measure the quality of approximation. It follows from the proof in Appendix A that

$$\frac{1}{2}\left\|W(t)W(t)^\top - V(t)V(t)^\top\right\|_F^2 - \frac{r-k}{2} = L_1(W(t), V(t)), \tag{17}$$

where $L_1(W(t), V(t))$ is a loss function defined in Theorem 1. We can alternatively rewrite

$$\frac{1}{2}\left\|W(t)W(t)^\top - V(t)V(t)^\top\right\|_F^2 - \frac{r-k}{2} = \sum_{i=1}^{k}\sin^2(\theta_i) \tag{18}$$

using the definition of principle angles $\theta_i$ between column spaces of matrices $W(t)$ and $V(t)$ (Björck & Golub, 1973). Given the later form, it is clear that small values of $\frac{1}{2}\left\|W(t)W(t)^\top - V(t)V(t)^\top\right\|_F^2 - \frac{r-k}{2}$ correspond to better aligned subspaces.

To demonstrate the main result of Theorem 2 we first prove a supplementary statement.

**Lemma 1.** *Let $A(t)$ be geodesic on $Gr(k_1, n)$, $A(t)^\top A(t) = I_{k_1}$. One can always construct geodesic $B(t)$, $B(t)^\top B(t) = I_{k_2}$ on $Gr(k_2, n), k_2 > k_1$ such that $\left\|\dot{B}(t)\right\|_F^2 \leq \left\|\dot{A}(t)\right\|_F^2$, where inequality is strict unless $\left\|\dot{A}(t)\right\|_F^2 \neq 0$.*

*Proof.* Since $A(t)$ is geodesic we can write $A(t) = A(0)Y\cos(\Sigma t)Y^\top + U\sin(\Sigma t)Y^\top$, where $U\sin(\Sigma t)Y^\top$ is singular value decomposition of $\dot{A}(0)$. Using orthogonality of $\dot{A}(0)$ and $A(0)$ we find $\left\|\dot{A}(t)\right\|_F^2 = \text{tr}(\Sigma)$. Without loss of generality we assume that $\Sigma_{11} \neq 0$. Consider

$$B(t) = \begin{pmatrix} A(0) & \Big| \\ & u_1 \\ & \Big| \end{pmatrix} \begin{pmatrix} y_1 y_1^\top + \sum_{i=2}^{k_1}\cos(\sigma_i t)y_i y_i^\top & 0 \\ 0 & 1 \end{pmatrix} + \sum_{i=2}^{k_1}\sin(\sigma_i t)u_i \begin{pmatrix} y_i \\ 0 \end{pmatrix}^\top$$

$$= \begin{pmatrix} A(0)Y\cos\left(\widetilde{\Sigma}t\right)Y^\top + U\sin\left(\widetilde{\Sigma}t\right)Y^\top & \Big| \\ & u_1 \\ & \Big| \end{pmatrix}, \tag{19}$$

where $y_i$ are columns of $Y$, $u_i$ are columns of $U$, $\sigma_i$ are diagonal elements of $\Sigma$, $\widetilde{\Sigma} = \Sigma - (\sigma_1 - 1)e_1 e_1^\top$, that is, $\widetilde{\Sigma}$ can be obtained from $\Sigma$ by replacing $\Sigma_{11} = \sigma_1$ by 1. Clearly $B(t)$ is geodesic and $\left\|\dot{B}(t)\right\|_F^2 = \left\|\dot{A}(t)\right\|_F^2 - \sigma_1 < \left\|\dot{A}(t)\right\|_F^2$. Next we show that principal angles between $A(t)$ and

$B(t)$ are all zero. To see this we observe that

$$
A^\top(t)B(t) = \left( Y\left(\cos\left(\Sigma t\right)\cos\left(\widetilde{\Sigma}t\right) + \sin\left(\Sigma t\right)\sin\left(\widetilde{\Sigma}t\right)\right)Y^\top \quad \Big| \quad y_1\sin(\sigma_1 t) \right)
$$

$$
= \left( Y\left(I - (1-\cos(\sigma_1 t))e_1 e_1^\top\right)Y^\top \quad \Big| \quad y_1\sin(\sigma_1 t) \right). \quad (20)
$$

From the identity above Frobenius norm reads

$$
\left\|A^\top(t)B(t)\right\|_F^2 = \sum_{i=1}^{k_1}\sin^2(\theta_i) = \mathrm{tr}\left(\cos^2(\sigma_1 t)y_1 y_1^\top + \sum_{i=2}^{k_1} y_i y_i^\top + y_1 y_1^\top\sin^2(\sigma_1 t)\right) = k_1, \quad (21)
$$

and we conclude that $\theta_i = 0$ for all $i = 1,\ldots,k_1$. $\qquad\square$

Lemma 1 also implies that for two such geodesics $\frac{1}{2}\left\|A(t)A(t)^\top - B(t)B(t)^\top\right\|_F^2 - \frac{k_2-k_1}{2} = 0$.

Now we are ready to show the main result of Theorem 2. We split interval of interest $t \in [0,T]$ on subintervals $[t_i, t_{i+1}]$ of length $\Delta t$. On each subinterval we consider three curves: (i) original continuously differentiable curve $V(t) \in \mathrm{Gr}(k,n)$, (ii) approximation of $V(t)$ by geodesic $Z(t) \in \mathrm{Gr}(k,n)$ passing through $V(t_i)$ with derivative $\dot{V}(t_i)$, (iii) embedding of $Z(t)$ by geodesic $W(t)$ on $\mathrm{Gr}(r,n), r > k$ selected as explained in Lemma 1. We start by showing that principle angles between $W(t)$ and $V(t)$ can be made arbitrary small

$$
\begin{aligned}
&\frac{1}{2}\left\|V(t)V(t)^\top - W(t)W(t)^\top\right\|_F^2 - \frac{r-k}{2} \\
&= \frac{1}{2}\left\|V(t)V(t)^\top - Z(t)Z(t)^\top + Z(t)Z(t)^\top - W(t)W(t)^\top\right\|_F^2 - \frac{r-k}{2} \\
&\leq \frac{1}{2}\left\|V(t)V(t)^\top - Z(t)Z(t)^\top\right\|_F^2 + \frac{1}{2}\left\|Z(t)Z(t)^\top - W(t)W(t)^\top\right\|_F^2 - \frac{r-k}{2} \\
&= \frac{1}{2}\left\|V(t)V(t)^\top - Z(t)Z(t)^\top\right\|_F^2.
\end{aligned}
\quad (22)
$$

Now we know that on each interval the distance between $V(t)$ and $W(t)$ is bounded by the distance from $V(t)$ to the geodesics that passes through $V(t_i)$ with speed $\dot{V}(t_i)$. Since interval is assumed to be small, we expand geodesic $Z(t)$ in Taylor series keeping terms proportional to $(\Delta t)^0$ and $\Delta t$ and for $V(t)$ we use Lagrange reminder $V(t) = V(t_i) + \dot{V}(\widetilde{t})(t - t_i), t \in [t_i, t_{i+1}], \widetilde{t} \in [t_i, t]$:

$$
\frac{1}{2}\left\|V(t)V(t)^\top - Z(t)Z(t)^\top\right\|_F^2 \simeq 2(t-t_i)^2\left\|\dot{V}(\widetilde{t}) - \dot{V}(t_i)\right\|_F^2. \quad (23)
$$

By assumption $V(t)$ is continuously differentiable, meaning the expression above can be made arbitrary small by selecting sufficiently small intervals $[t_i, t_{i+1}]$.

To show that derivative of $W(t)$ can be made smaller than $V(t)$ observe that $\left\|\dot{W}(t)\right\|_F^2 < \left\|\dot{Z}(t)\right\|_F^2$ on each subinterval where $\left\|\dot{Z}(t)\right\|_F^2 \neq 0$. Since $\left\|\dot{Z}(t)\right\|_F^2 = \left\|\dot{Z}(t_i)\right\|_F^2 = \left\|\dot{V}(t_i)\right\|_F^2$ and $\dot{V}(t)$ is continuous function, we, again, can select sufficiently small intervals such that deviation of $\left\|\dot{V}(t)\right\|_F^2$ from $\left\|\dot{V}(t_i)\right\|_F^2$ on each interval is small enough for $\left\|\dot{W}(t)\right\|_F^2 < \left\|\dot{V}(t)\right\|_F^2$ to hold.

## C  SUBSPACE EMBEDDING EXAMPLE

The proof of Theorem 2 is constructive, meaning we can compute $W(t)$ given $V(t)$ and $\dot{V}(t)$ or its estimation. We select

$$
V(t) = \gamma_1(t) = \begin{pmatrix} \sin(\theta(t))\sin(\phi(t)) \\ \cos(\theta(t))\sin(\phi(t)) \\ \cos(\phi(t)) \end{pmatrix}, \theta(t) = 7\pi\cos(2\pi t), \phi(t) = \pi/2 + \pi/4\cos(2\pi t). \quad (24)
$$

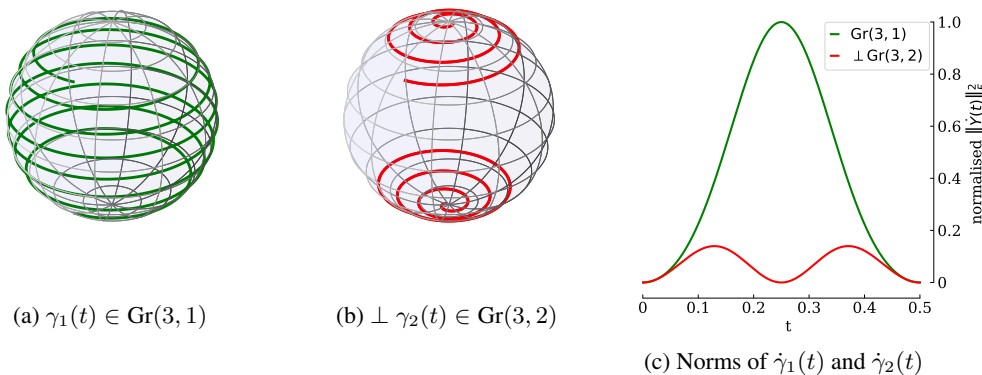

(a) $\gamma_1(t) \in \mathrm{Gr}(3, 1)$      (b) $\perp \gamma_2(t) \in \mathrm{Gr}(3, 2)$      (c) Norms of $\hat{\gamma}_1(t)$ and $\hat{\gamma}_2(t)$

Figure 4: Example of subspace embedding detailed in Appendix C.

Curve $\gamma_1(t) \in \mathrm{Gr}(1, 3)$ is illustrated in Figure 4a. We next estimate derivatives by splitting $t$ on a set of subintervals and taking logarithm on each interval. This derivative is used as explained in Lemma 1 to define $W(t) \in \mathrm{Gr}(2, 3)$. Since $\frac{1}{2}\left\|\dot{P}_W\right\|_F^2 = \left\|\dot{W}(t)\right\|_F^2 = \frac{1}{2}\left\|\frac{d}{dt}(I - P_W)\right\|_F^2$ we plot $\gamma_2(t) \perp W(t)$ in Figure 4b. Curve $\gamma_2(t)$ appears to be discontinuous, but actually it is continuous owning to $\mathbb{Z}_2$ symmetry of compact Stiefel manifold $\mathrm{St}(1, 3)$. Norms of derivative are compared in Figure 4c: curve $\gamma_2$ is manifestly smoother than $\gamma_1$.

## D    PROOF OF THEOREM 3

In the proof we will write $F$ ad $G$ in place of $F_k$ and $G_k$ assuming that $k$ is fixed and the value of $k$ is evident from the context.

### D.1    PARTS 1. AND 2.

We order eigenvectors in the increase of eigenvalue $E(i_1, \ldots, i_D) := \lambda_{i_1, \ldots, i_D} = \sum_{j=1}^D a_j i_j^2$, which we will also call energy in this section. To understand how eigenvectors and subspaces are selected for different coefficients $a_1, \ldots, a_D$ we introduce continuous relaxation of energy $E(z_1, \ldots, z_D) = \sum_{j=1}^D a_j z_j^2$, where $z_j \in \mathbb{R}_+$. In continuous form, surfaces with constant energies are (hyper)ellipsoids of dimension $D - 1$, so the process of selecting $k$-th eigenvector or constructing subspace of dimension $k$ can be understood through the following informal algorithm:

1. Select $a_1, \ldots, a_D$ and $c = 0$.

2. Gradually increase $c$ and track ellipsoid $\sum_{j=1}^D a_j z_j^2 = c$.

3. While increasing $c$ add each standard positive lattice point (point with positive integer coordinates) that fall inside the ellipsoids.

4. The order at which lattice points cross an inflating ellipsoid define which eigenvector appears on position $k$ and which vectors form eigenspace of dimension $k$.

To illustrate this process, consider $E(z_1, \ldots, z_D) = a_1 z_1^2 + a_2 z_2^2$, where $a_2 \gg a_1$. If we follow procedure outlined above we will see that first lattice points encountered are $(1, 1), (2, 1), (3, 1), (4, 1), \ldots$. So for considered $a_1, a_2$ the subspace of first 3 eigenvectors is a span of $\phi_{1,1}, \phi_{2,1}, \phi_{3,1}$, and the eigenvector that appears on position 3 is $\phi_3$. To describe the map from $a_1, \ldots, a_D$ to $\phi_k$ or $V_k$, this procedure needs to be repeated for all possible positive values of real coefficients $a_1, \ldots, a_D$.

From the algorithm above one can deduce that for given $a_1, \ldots, a_D$ the first time eigenvector with indices $i_1, \ldots, i_D$ appears in the sequence of eigenvectors is the first time ellipsoid crosses $i_1, \ldots, i_D$. The position $k$ of this eigenvector will be proportional to the normalised volume of the ellipsoid $V_e(a_1, \ldots, a_D)/V_s$, where $V_e(a_1, \ldots, a_D)$ is a volume of $D$ dimensional ellipsoid with semi-axes $a_1, \ldots, a_D$ and $V_s$ is a volume of $D$ dimensional sphere with radius 1.

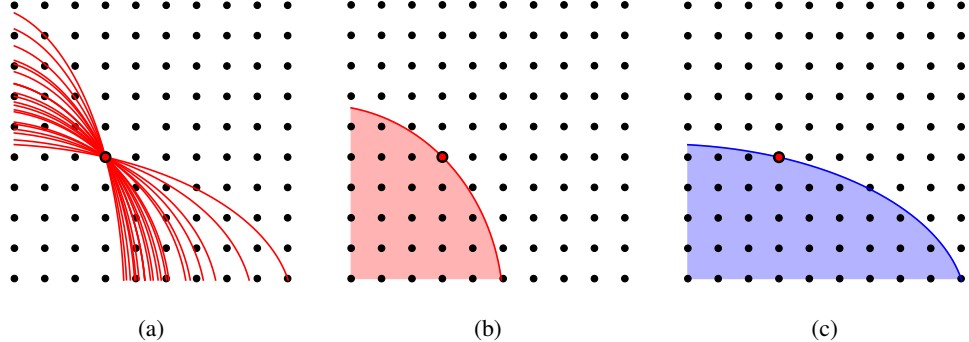

(a)                                    (b)                                    (c)

Figure 5: (a) Parametric family of ellipsoids passing through point $(4, 5)$. (b) Ellipsoid of minimal volume passing through $(4, 5)$. Note that the number of standard lattice points inside is approximately $4 \times 5 = 20$, the error of approximation (5 in this case) is asymptotically small for ellipsoids of large volume. (c) Example of non-minimal ellipsoid passing through $(4, 5)$. In the non-minimal case, the number of standard lattice points inside an ellipsoid passing through a given point can be made arbitrarily large.

The first immediate consequence is that $a_1, \ldots, a_D$ is a piecewise constant function. Indeed, it is clear $a_1, \ldots, a_D$ can always be perturbed with no change in filling order and the change of $-1 < V_e(a_1, \ldots, a_D)/V_s < 1$, so the eigenvector on position $k$ does not change. That proves the first part of the first statement. Next, we need to show that the set of all possible eigenvectors on position $k$ is finite.

To see that, we answer the following question: what is the minimal number of lattice points one ought to cover with an ellipsoid to reach a given lattice point $i_1, \ldots, i_D$? For example, point $(1, 1)$ is always reached first. On the other hand, point $(2, 1)$ can be reached arbitrarily late, because one may consider ellipsoids with arbitrary large semi-axis along the second dimension. The position of the point $i_1, \ldots, i_D$ is known to be the ratio of volumes, so we need to find an ellipsoid with minimal volume that passes through $i_1, \ldots, i_D$. A parametric family of ellipsoids in question and its volume are

$$\sum_{i=1}^{D} z_i^2 \frac{a_i^2}{\left(\sum_{k=1}^{D} a_k^2 i_k^2\right)^{\frac{1}{2}}} = 1, \quad V_e = \frac{\pi^{\frac{D}{2}}}{\Gamma\left(\frac{D}{2}+1\right)} \frac{\left(\sum_{k=1}^{D} a_k^2 i_k^2\right)^{\frac{D}{2}}}{\left(\prod_{j=1}^{D} a_j^2\right)^{\frac{1}{2}}}. \tag{25}$$

See Figure 5a for example of a parametric family in $D = 2$ passing through lattice point $(4, 5)$. In the expression above we used $a_i^2$ to remove positivity constraints. To find minimal volume we take derivative with respect to $a_k$

$$\frac{\partial V_e}{\partial a_k} = 0 \Rightarrow D a_k^2 i_k^2 - \sum_{j=1}^{D} a_j^2 i_j^2 = 0. \tag{26}$$

To find $a_i^2$ we need to compute the nullspace of the linear operator above. It is easy to see that the solution is $a_k^2 = \frac{\alpha}{i_k^2}$ for arbitrary $\alpha \in \mathbb{R}$. The example of minimal ellipsoid appears in Figure 5b. The volume does nod depend on multiplicative constant so we take $\alpha = 1$ and obtain normalised minimal volume

$$\frac{\min V_e(i_1, \ldots, i_D)}{V_s} = \prod_{j=1}^{D} i_j. \tag{27}$$

From the considerations above we can conclude that: (i) eigenvector $i_1, \ldots, i_D$ can not appear on position $k$ unless $\prod_{j=1}^{D} i_j < k$, (ii) if eigenvector $i_1, \ldots, i_D$, excluding $1, \ldots, 1$, appears on position $k$, it can also appear on any position $l > k$. Statement (ii) is correct because parametric family of ellipsoids passing through $i_1, \ldots, i_D$ contain ellipsoids of arbitrary large volumes unless $i_j = 1$ for all $j = 1, \ldots, D$. The example of a non-minimal ellipsoid is in Figure 5c.

Statement (ii) directly leads to point 2. of Theorem 3. Indeed, since any eigenvector appeared on position $k$ can reappear on arbitrary position $l > k$, the number of unique vectors that can form low-energy subspace of dimension $l$ is the number of eigenvectors on position $l$ plus eigenvector $1, \ldots, 1$. That finished the proof of point 1. and 2. of Theorem 3.

Note, that the validity of most of the statements in this section is based on assumptions that we can use continuous relaxation on the problem. In particular, we assumed that the position of the eigenvector is proportional to the volume of the ellipsoid. Of course in completely discrete formulation this is not strictly the case, but since all statements on the number of eigenvectors and eigenspaces are asymptotic, they will remain valid.

### D.2 PART 3.

From the previous section we know the minimal position eigenvector $i_1, \ldots, i_D$ can appear at. Besides that we know that once $i_1, \ldots, i_D$ is unlocked, it can appear on all positions $l > k$. Give that, the number of eigenvectors on position $k$ reads

$$\#_F(k, D) = \sum_{i_1=1}^{\infty} \cdots \sum_{i_D=1}^{\infty} \mathrm{Ind} \left[ \prod_{j=1}^{D} i_j \leq k \right], \tag{28}$$

where $\mathrm{Ind}[\cdot]$ is an indicator function. We are interested in asymptotic expansion for large $k$ and fixed $D$, so the sums above can be approximated by the Euler–Maclaurin formula.

To find asymptotic expansion we will derive recurrence relations for $\#_F(k, D)$. We start by introducing a slightly modified function

$$\widetilde{\#}_F(\alpha, D) = \frac{1}{\alpha} \sum_{i_1=1}^{\infty} \cdots \sum_{i_D=1}^{\infty} \mathrm{Ind} \left[ \prod_{j=1}^{D} i_j \leq \alpha \right]. \tag{29}$$

Clearly $\#_F(k, D) = k\widetilde{\#}_F(k, D)$ so if we know how to compute $\widetilde{\#}_F(\alpha, D)$, we can recover $\#_F(k, D)$. For $D = 2$ with the help of Euler–Maclaurin formula we obtain

$$\widetilde{\#}_F(\alpha, 2) = \frac{1}{\alpha} \sum_{i_1=1}^{\infty} \mathrm{Ind}\left[i_1 \leq \alpha\right] \sum_{i_2=1}^{\frac{\alpha}{i_1}} 1 = \frac{1}{\alpha} \sum_{i_1=1}^{\alpha} \frac{\alpha}{i_1} \sim \frac{1}{\alpha} \left( \int_1^{\alpha} dx \frac{\alpha}{x} + \frac{\alpha+1}{2} \right) = \log\alpha + \frac{1}{2} + \frac{1}{2\alpha}. \tag{30}$$

Next we find recurrence relation

$$\widetilde{\#}_F(\alpha, D+1) = \frac{1}{\alpha} \sum_{i_{D+1}=1}^{\alpha} \frac{1}{i_{D+1}} \sum_{i_D=1}^{\frac{\alpha}{i_{D+1}}} \frac{1}{i_D} \cdots \sum_{i_D=1}^{\frac{\alpha}{i_{D+1} i_D \cdots i_2}} \frac{1}{i_2}$$

$$= \sum_{i_{D+1}=1}^{\alpha} \frac{\widetilde{\#}_F\left(\frac{\alpha}{i_{D+1}}, D\right)}{i_{D+1}} \sim \int_1^{\alpha} dx \frac{\widetilde{\#}\left(\frac{\alpha}{x}, D\right)}{x} + \frac{1}{2} \left( \frac{\widetilde{\#}(1, D)}{\alpha} + \widetilde{\#}(\alpha, D) \right). \tag{31}$$

It is not hard to show that, starting from $D = 2$, recurrence relation can only produce three type of terms: $\log^p(\alpha)$, constant term $c$, $\frac{1}{\alpha}$. This can be seen as follows

$$\log^p(\alpha) \to \int_1^{\alpha} \frac{\log^p(\alpha/x)}{x} + \frac{1}{2} \log^p(\alpha) = \frac{1}{p+1} \log^{p+1}(\alpha) + \frac{1}{2} \log^p(\alpha),$$

$$c \to \int_1^{\alpha} dx \frac{c}{x} + \frac{c}{2} \left( \frac{1}{\alpha} + 1 \right) = c\log(k) + \frac{c}{2\alpha} + \frac{c}{2}, \tag{32}$$

$$\frac{1}{\alpha} \to \int_1^{\alpha} \frac{dx}{\alpha} + \frac{1}{\alpha} = 1.$$

Given that, starting from $\widetilde{\#}_F(\alpha, 2)$ and applying recurrence relations $D - 2$ times we obtain leading term

$$\widetilde{\#}_F(\alpha, D) \sim \frac{1}{(D-1)!} \log(\alpha)^{D-1} \Rightarrow \#_F(k, D) \sim \frac{k}{(D-1)!} \log(k)^{D-1}, \tag{33}$$

where last identity follows from the definition of $\widetilde{\#}_F(\alpha, D)$.

### D.3 PART 4.

We cannot apply the Euler–Maclaurin formula when $k$ is fixed and $D$ is large. To count states under specified conditions we will use factorisation on prime numbers. For positive integer $p$ we can write

$$p = q_1(p)^{a_1(p)} \cdots q_{m_p}(p)^{a_{m_p}(p)}, \tag{34}$$

where $q_i(p)$ are prime factors and $a_i(p)$ are their multiplicities. Given this factorisation we can find the number of ways positive integer $p$ can be represented as products of $D$ positive integers. All products of $D$ integers correspond to some rearrangement of products in the factorisation of prime factors. The number of such rearrangements is

$$\tau(p, D) = \prod_{r=1}^{m_p} \frac{(a_r(p) + D - 1)!}{(D-1)! \, a_r(p)!}. \tag{35}$$

This expression is easy to understand if one considers forming the product of $D$ numbers by distributing $q_r(p)$ to selected $a_r(p)$ among $D$ factors for each prime factor $q_r(p), r = 1, \ldots, m_p$.

From the expression above, the number of states on position $k$ reads

$$\#(k, D) = \sum_{p=1}^{D} \tau(p, D). \tag{36}$$

If $D$ is large

$$\tau(p, D) = \prod_{r=1}^{m_p} \frac{(a_r(p) + D - 1)!}{(D-1)! \, a_r(p)!} \sim \prod_{r=1}^{m_p} \frac{D^{a_r(p)}}{a_r(p)!} = \frac{D^{\sum_{r=1}^{m_p} a_r(p)}}{\prod_{r=1}^{m_p} a_r(p)!} = \frac{D^{\Omega(p)}}{\prod_{r=1}^{m_p} a_r(p)!}, \tag{37}$$

where $\Omega(p)$ is the prime (big) omega function.

Leading asymptotic expansion of the sum is the fastest growing term

$$\#(k, D) \sim D^{\max_{p \le k} \Omega(p)} \sum_{l \in \arg\max_{p \le k} \Omega(p)} \frac{1}{\prod_{r=1}^{m_l} a_r(l)!}. \tag{38}$$

In the main body of the text we provide a simplified upper bound of this asymptotic expansion. It can be derived using two upper bounds. First, prime omega function can be bounded from above

$$p = q_1(p)^{a_1(p)} \cdots q_{m_p}(p)^{a_{m_p}(p)} \ge 2^{a_1(p) + \cdots + a_{m_p}(p)} = 2^{\Omega(p)} \to \Omega(p) \le \log_2(p). \tag{39}$$

Next, the remaining sum can be bounded from above

$$\sum_{l \in \arg\max_{p \le k} \Omega(p)} \frac{1}{\prod_{r=1}^{m_l} a_r(l)!} \le \sum_{l=1}^{k} \frac{1}{\prod_{r=1}^{m_l} a_r(l)!} \le k. \tag{40}$$

These two upper bound combined gives us

$$\#(k, D) \le k D^{\log_2(k)}. \tag{41}$$

### D.4 PART 5.

We were unable to compute exact asymptotic expansions for the number of subspaces, so our strategy in this and next section will be to derive sufficiently strong lower bound by counting selected ways subspaces can be formed.

Consider $D = 2$ and $k = 4$. Since surfaces with constant energies are ellipsoids, we can select $a_1 = 1$ large $a_2$ and by gradual decrease of $a_2$ we will observe three distinct subspaces:

$$\{\phi_{1,1}, \phi_{1,2}, \phi_{1,3}, \phi_{1,4}\} \to \{\phi_{2,1}, \phi_{1,1}, \phi_{1,2}, \phi_{1,3}\} \to \{\phi_{2,1}, \phi_{2,2}, \phi_{1,1}, \phi_{1,2}\}. \tag{42}$$

Similarly, starting from $a_2 = 1$ and large $a_1$ decrease of $a_1$ lead to the sequence of subspaces

$$\{\phi_{1,1}, \phi_{2,1}, \phi_{3,1}, \phi_{4,1}\} \to \{\phi_{1,2}, \phi_{1,1}, \phi_{2,1}, \phi_{3,1}\} \to \{\phi_{1,2}, \phi_{2,2}, \phi_{1,1}, \phi_{2,1}\}. \tag{43}$$

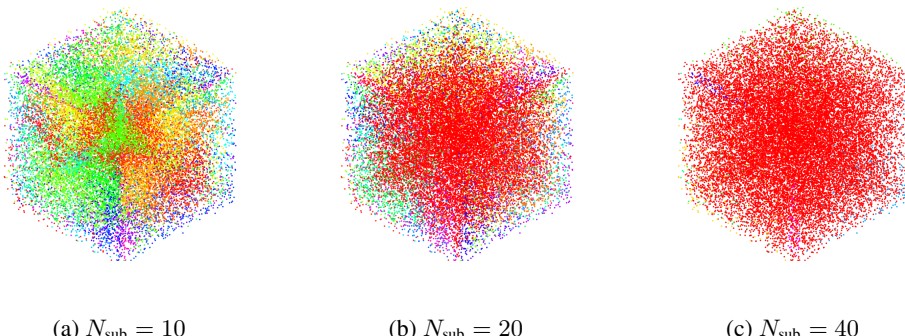

(a) $N_{\text{sub}} = 10$         (b) $N_{\text{sub}} = 20$         (c) $N_{\text{sub}} = 40$

Figure 6: Illustration of simple greedy subspace embedding technique for elliptic eigenproblem with constant coefficients. See Appendix E for details.

Sequences above can be understood as systematic fillings of lattice points along second and first dimensions. First sequence corresponds to filling $(0, 4)$, $(1, 3)$, $(2, 2)$ points, and second to $(4, 0)$, $(3, 1)$, $(2, 2)$. In $D = 2$ the number of distinct subspaces $V_k$ constructed in this way equals the number of unordered pairs $k_1, k_2 \geq 0$ such that $k_1 + k_2 = k$.

In arbitrary $D$ similarly constructed states can be counted as the number of unordered tuples $(k_1, \ldots, k_D)$ of non-negative integers such that $\sum_{i=1}^{D} k_i = k$. This is a standard counting problem with the answer

$$\frac{(k + D - 1)!}{(D - 1)!k!} \sim \frac{k^{D-1}}{(D - 1)!}. \tag{44}$$

This provides a lower bound on asymptotic expansion because our way to select subspaces is not exhaustive.

### D.5 PART 6.

For fixed $k$ and large $D$ we consider eigenvectors with indices $(1, \ldots, 1)$, $(1, 2, 1, \ldots, 1)$, $\ldots$, $(1, \ldots, 1, 2, 1)$. It is clear that the first eigenvector has indices $(1, \ldots, 1)$ and the rest of them can appear in arbitrary order. This gives us at least $\frac{D(D-1)\cdots(D-k+1)}{(k-1)!} \sim \frac{1}{(k-1)!} D^{k-1}$ subspaces.

## E MONTE CARLO EXPERIMENTS FOR EIGENPROBLEM WITH CONSTANT COEFFICIENTS

To illustrate consequences of Theorem 3 we perform simple Monte Carlo experiment. For the case $D = 3$, we generate $a_1, a_2, a_3$ from uniform distribution on $[0, 1]$ repeatedly and record distinct subspaces of dimension $k = 10$. We select unique colour for each subspace and draw them for each point $a_1, a_2, a_3$ on the plane perpendicular to $\begin{pmatrix} 1 & 1 & 1 \end{pmatrix}^{\top}$. This illustration appears in Figure 6a. Theorem 3 suggest that the large number of distinct subspaces is a result of selection of $k$ vectors among a small number of candidates. This suggests we may decrease complexity of the function from coefficients to subspaces by predicting excessive number of vectors. In our experiments with neural networks this redundant mapping is learned, here we build the mapping with simple greedy strategy. In place of function $a_1, a_2, a_3 \to V_{10}$ we consider $a_1, a_2, a_3 \to V_{10} \cup \{v_1, \ldots, v_k\}$ where $v_1, \ldots, v_k$ are first $k$ most abundant eigenvectors. By appending additional vectors we decrease the number of distinct subspaces. The result of this greedy simplification appear in Figure 6b with 10 additional vectors and in Figure 6c with 20 additional vectors. As we see the number of distinct subspaces rapidly decreasing.

## F  Details on numerical experiments for eigenproblems

### F.1  Datasets

We generated two datasets for $D = 2$ elliptic eigenproblem with uniform Dirichlet boundary conditions

$$\text{div } k \cdot \text{grad } \phi_i = \lambda_i \phi_i, \ \|\phi_i\|_2 = 1, \ x \in [0,1]^2. \tag{45}$$

For both datasets we used uniform grid $100 \times 100$ and finite-difference discretisation. Components of diffusion coefficients were generated from the same distribution for both datasets. Diffusion coefficient is generated as follows:

1. Gaussian random field $\psi$ is generated from $\mathcal{N}\left(0, (\text{id} - \gamma \Delta)^r\right), \gamma = \frac{1}{20\pi}, r = \frac{1}{2}$.
2. Diffusion coefficient is computed as $a = \alpha + (\beta - \alpha)\left(\tanh\left(s\psi\right) + 1\right)/2$ with $\alpha = 1$, $\beta = 50, s = 1$.

For one $D = 2$ dataset $k_1 = k_2$ and for another $k_1 \neq k_2$ but both are i.i.d. random fields generated as described above. In the main text only results for $k_1 = k_2$ are reported.

For $D = 3$ elliptic eigenproblem we use setup analogous to $D = 2$ but grid of size $30 \times 30 \times 30$ and $k_1 = k_2 = k_3$ generated the same way as explained above with parameters $\gamma = \frac{1}{100}, r = \frac{3}{2}$, $\alpha = 50, \beta = 1, s = 2$.

For QM problems datasets are defined by distributions for potential functions.

For $D = 1$ we use

$$V(r) = d\left(1 - \exp\left(-\frac{\frac{r}{r_e} - 1}{\frac{r}{r_e} + 1}p\left(r\right)\right)\right)^2, p(r) = \begin{cases} q_1\left(\frac{r}{r_e}\right), \ r < r_e \\ q_2\left(\frac{r}{r_e}\right), \ r \geq r_e \end{cases}, \tag{46}$$

where

$$q_1(x) = \left(1 - \frac{x-1}{x+1}\right)\widetilde{q}_1(x) + c\frac{x-1}{x+1}, \tag{47}$$

and $\widetilde{q}_1(x)$ is a polynomial of degree deg. Polynomial $q_2(x)$ has the same form.

In $D = 1$ dataset is by selecting uniform grid with 100 points on the interval $[0, 10]$, $r_e$ is sampled from uniform distribution on the interval $[1, 8]$, $d$ is sampled from uniform distribution on the interval $[10, 40]$, both $q_1$ and $q_2$ has order 10, for $q_1$ all coefficients (including $c$) are sampled from uniform distribution on $[0, 5]$, for $q_2$ coefficients of $\widetilde{q}_2$ are sampled from uniform distribution on $[0, 10]$ and $c$ is sampled from uniform distribution on the interval $[1, 11]$.

For $D = 2$

$$V(x,y) = V_1\left(\sqrt{(x - cu)^2 + (y - cv)^2}\right) + V_2\left(\sqrt{(x + cu)^2 + (y + cv)^2}\right), \tag{48}$$

where $u, v$ are component of random normalised vector, $c = \sqrt{2}r_e$. Potentials $V_1$ and $V_2$ are i.i.d. with parameters: order of polynomial is 2, $r_e$ is uniformly distributed on $[1, 5]$, $d$ is uniformly distributed on $[10, 40]$, coefficients of $\widetilde{q}$ are sampled from uniform distribution on $[0, 3]$ and $c$ is sampled from uniform distribution on $[10, 13]$. To discretise the problem we use finite difference and uniform $100 \times 100$ grid on $[-7, 7]^2$.

### F.2  Architectures and training

In all cases we used FFNO architecture, with GELU activation functions, that is completely specified by: number of layers $N_{\text{layers}}$, numbers of features in hidden layer $N_{\text{features}}$, number of Fourier modes in spectral convolution $N_{\text{modes}}$. Since all our loss functions are scale-invariant, the output of FFNO architecture was normalised.

We use Lion optimiser (Chen et al., 2023), with weight decay. Parameters of the optimiser are learning rate lr, rate decay factor $\gamma_{\text{decay}}$ and number of transition steps $N_{\text{decay}}$.

For $D = 2$, eigenvalue and QM problems, and also $D = 1$ QM problem we perform grid search with parameters: $N_{\text{layers}} \in [3, 4, 5]$, $N_{\text{features}} = 64$, $N_{\text{modes}} \in [10, 14, 16]$, lr $\in [10^{-3}, 10^{-4}]$, $\gamma_{\text{decay}} = 0.5$,

$N_{\text{decay}} \in [100, 200]$, batch size was fixed to 100, number of train samples is 4000, number of test samples is 1000. Architecture is training to approximate subspace spanned by first 10 eigenvectors. Number of epoch is 1000. When architecture is trained to predict individual eigenvectors, the same grid search applies.

Since architectures used are neural operators, we were able to speed up grid search by a factor of 7 by first training architecture on coarse grid $32 \times 32$, and then retraining 3 most accurate architectures (according to the results of grid search on the coarse grid) of the fine $100 \times 100$ grid.

For $D = 3$ grid search is not practical, so we select $N_{\text{layers}} = 4$, $N_{\text{features}} = 128$, $N_{\text{modes}} = 16$, lr $= 10^{-3}$, $\gamma_{\text{decay}} = 0.5$, $N_{\text{decay}} = 100$. The size of the train set is 800, the size of test set is 200. The number of epochs is 1000. Architecture is training to approximate subspace spanned by first 3 eigenvectors.

### F.3 ADDITIONAL RESULTS FOR ELLIPTIC EIGENPROBLEMS

Additional results are available in Table 3, Table 4, Table 5. Results in brackets indicate worst and best observed result among three best grid search runs.

Table 3: Comparison of $L_1(A, B)$ and $L_2(A, B; z)$ loss functions for $k_1 = k_2$

| | $L_1(A, B)$ | | | $L_2(A, B; z)$ | | |
|---|---|---|---|---|---|---|
| $N_{\text{sub}}$ | $E_{\text{train}}$ | $E_{\text{test}}$ | $t_{\text{train}}$, s | $E_{\text{train}}$ | $E_{\text{test}}$ | $t_{\text{train}}$, s |
| 10 | $[0.216, 0.235]$ | $[0.292, 0.314]$ | $4124 \pm 443$ | $[0.244, 0.252]$ | $[0.296, 0.302]$ | $3895 \pm 524$ |
| 20 | $[0.038, 0.048]$ | $[0.046, 0.052]$ | $5962 \pm 93$ | $[0.046, 0.052]$ | $[0.049, 0.058]$ | $4074 \pm 215$ |
| 30 | $[0.024, 0.029]$ | $[0.028, 0.033]$ | $7902 \pm 85$ | $[0.026, 0.033]$ | $[0.029, 0.037]$ | $3973 \pm 515$ |
| 40 | $[0.018, 0.025]$ | $[0.021, 0.029]$ | $10842 \pm 570$ | $[0.017, 0.024]$ | $[0.02, 0.027]$ | $4270 \pm 95$ |

Table 4: Comparison of $L_1(A, B)$ and $L_2(A, B; z)$ loss functions for $k_1 \neq k_2$

| | $L_1(A, B)$ | | | $L_2(A, B; z)$ | | |
|---|---|---|---|---|---|---|
| $N_{\text{sub}}$ | $E_{\text{train}}$ | $E_{\text{test}}$ | $t_{\text{train}}$, s | $E_{\text{train}}$ | $E_{\text{test}}$ | $t_{\text{train}}$, s |
| 10 | $[0.305, 0.377]$ | $[0.407, 0.421]$ | $4035 \pm 295$ | $[0.312, 0.335]$ | $[0.386, 0.389]$ | $3661 \pm 308$ |
| 20 | $[0.066, 0.089]$ | $[0.09, 0.105]$ | $5817 \pm 219$ | $[0.092, 0.092]$ | $[0.103, 0.103]$ | $4262 \pm 0$ |
| 30 | $[0.05, 0.05]$ | $[0.063, 0.063]$ | $7966 \pm 84$ | $[0.042, 0.05]$ | $[0.052, 0.059]$ | $3991 \pm 168$ |
| 40 | $[0.035, 0.036]$ | $[0.045, 0.047]$ | $11019 \pm 252$ | $[0.034, 0.038]$ | $[0.041, 0.046]$ | $4238 \pm 112$ |

Table 5: $\mathbb{Z}_2$-adjusted $L_2$ loss.

| | $k_1 = k_2$ | | | $k_1 \neq k_2$ | | |
|---|---|---|---|---|---|---|
| $N_{\text{eig}}$ | $E_{\text{train}}$ | $E_{\text{test}}$ | $t_{\text{train}}$, s | $E_{\text{train}}$ | $E_{\text{test}}$ | $t_{\text{train}}$, s |
| 0 | $[0.009, 0.012]$ | $[0.036, 0.038]$ | $3666 \pm 476$ | $[0.014, 0.028]$ | $[0.068, 0.07]$ | $3280 \pm 60$ |
| 1 | $[0.038, 0.042]$ | $[0.158, 0.165]$ | $3312 \pm 77$ | $[0.031, 0.033]$ | $[0.196, 0.218]$ | $3956 \pm 162$ |
| 2 | $[0.046, 0.048]$ | $[0.359, 0.373]$ | $4168 \pm 82$ | $[0.045, 0.053]$ | $[0.553, 0.563]$ | $4140 \pm 87$ |
| 3 | $[0.046, 0.057]$ | $[0.541, 0.555]$ | $4168 \pm 81$ | $[0.054, 0.066]$ | $[0.747, 0.779]$ | $4101 \pm 87$ |
| 4 | $[0.068, 0.084]$ | $[0.754, 0.769]$ | $4007 \pm 204$ | $[0.072, 0.078]$ | $[0.945, 0.97]$ | $4101 \pm 87$ |
| 5 | $[0.073, 0.075]$ | $[0.897, 0.905]$ | $4041 \pm 264$ | $[0.084, 0.094]$ | $[1.087, 1.098]$ | $4090 \pm 83$ |

### F.4 GENERALISATION TO DIFFERENT GRID SIZE

For all subspace regression problems we use FFNO. Since FFNO is neural operator it should be discretisation agnostic. Here we report results for model trained on grid 100 on $D = 1$ quantum mechanics eigenproblem and tested on grids of higher resolution. For each grid we generated new test set from the same distribution as specified in Appendix F.1.

The results are available in Table 6. We see approximately linear increase of relative error with resolution. Owning to good initial accuracy on grid $N_x = 100$ the relative error remains under $10\%$ for grid with $N_x = 500$.

### F.5 SMOOTHNESS OF NEURAL NETWORKS TRAINED WITH SUBSPACE EMBEDDING TECHNIQUE

To empirically measure smoothness of learned map, we introduce several "smoothness indicators":

1. Taylor indicator

$$T[f_1, f_2; l] = \frac{\left\| \mathcal{N}(f_1 + lf_2) - \mathcal{N}(f_1) - \frac{d}{dl}\mathcal{N}(f_1 + lf_2)\big|_{l=0}\, l \right\|_2}{\left\| \mathcal{N}(f_1 + lf_2) \right\|_2}. \qquad (49)$$

Taylor indicator is a relative error of linear model. One expect that: (i) unless $\mathcal{N}$ is a linear function, when $l$ increases relative error also increases; (ii) when smoothness increases Taylor indicator decreases.

2. Average cosine

$$C[f_1, f_2; l] = \frac{1}{D} \sum_{i=1}^{D} \cos_i(\mathcal{N}(f_1 + lf_2), t_1). \qquad (50)$$

where $\cos_i(A, B)$ are cosines of principle angles (Björck & Golub, 1973) and $t_1$ is target at point $f_1$. For smoother maps average cosine increases until it reaches maximal value of 1.

3. Frobenius norm of the directional derivative

$$F[f_1, f_2; l] = \frac{1}{D} \left\| \frac{d}{dl}\mathcal{N}(f_1 + lf_2)\Big|_{l=0} \right\|_F, \qquad (51)$$

where $D$ is the subspace size. The magnitude of the directional derivative is computed by automatic differentiation and it is expected to decrease when smoothness increases.

Each indicator depends on two features $f_1$ and $f_2$ and real number $l \in [0, 1]$. Results are reported for neural network trained to predict eigenspaces for elliptic eigenproblem with $k_1 = k_2$ and $k_1 \neq k_2$ (see Appendix F.1 for description) with loss function $L_2(A, B; z)$. For each indicator we provide values for several $l$ averaged over 1000 randomly selected feature pairs $f_1$, $f_2$. Results are reported in Table 7, Table 8, Table 9.

All indicators clearly demonstrate the improve in smoothness when the size of embedding $N_{\text{subspace}}$ increases. Results for average cosine indicator (50)

Table 7: Frobenius norm of the directional derivative smoothness indicator (51).

| $N_{\text{subspace}}$ | $k_1 = k_2$ | $k_1 \neq k_2$ |
|---|---|---|
| 10 | 10.73 | 8.26 |
| 20 | 9.72 | 8.15 |
| 30 | 8.78 | 8.14 |
| 40 | 8.26 | 6.88 |

Table 6: Network is trained on resolution $N_x = 100$ for $D = 1$ QM problem and evaluated on grids with increased resolution.

| $N_x$ | 100 | 150 | 200 | 250 | 300 | 350 | 400 | 450 | 500 |
|---|---|---|---|---|---|---|---|---|---|
| test error, % | 0.83 | 1.37 | 2.12 | 2.95 | 3.96 | 4.32 | 5.33 | 5.32 | 6.79 |

indicate that learned mapping effectively average
information about subspaces for distinct features. It becomes especially clear if one compares results for $l = 1.0$ with average cosines computed between targets $\frac{1}{D} \sum_{i=1}^{D} \cos_i(t_1, t_2)$: for $k_1 = k_2$ average cosine is $0.51$; for $k_1 \neq k_2$ average cosine is $0.53$.

Table 8: Taylor smoothness indicator (49).

| $N_{\text{subspace}}$ | $k_1 = k_2$ | | | | $k_1 \neq k_2$ | | | |
|---|---|---|---|---|---|---|---|---|
| | $l = 0.25$ | $l = 0.5$ | $l = 0.75$ | $l = 1.0$ | $l = 0.25$ | $l = 0.5$ | $l = 0.75$ | $l = 1.0$ |
| 10 | 2.41 | 5.1 | 7.48 | 10.71 | 1.8 | 3.87 | 5.99 | 8.21 |
| 20 | 2.12 | 4.57 | 7.05 | 9.68 | 1.72 | 3.77 | 5.86 | 8.08 |
| 30 | 1.88 | 4.09 | 6.34 | 8.73 | 1.71 | 3.77 | 5.87 | 8.08 |
| 40 | 1.76 | 3.84 | 5.98 | 8.22 | 1.42 | 3.15 | 4.94 | 6.8 |

Table 9: Average cosine indicator (50).

| $N_{\text{subspace}}$ | $k_1 = k_2$ | | | | $k_1 \neq k_2$ | | | |
|---|---|---|---|---|---|---|---|---|
| | $l = 0.25$ | $l = 0.5$ | $l = 0.75$ | $l = 1.0$ | $l = 0.25$ | $l = 0.5$ | $l = 0.75$ | $l = 1.0$ |
| 10 | 0.81 | 0.74 | 0.67 | 0.51 | 0.82 | 0.74 | 0.67 | 0.53 |
| 20 | 0.94 | 0.88 | 0.82 | 0.68 | 0.94 | 0.88 | 0.83 | 0.72 |
| 30 | 0.97 | 0.93 | 0.88 | 0.78 | 0.97 | 0.93 | 0.89 | 0.81 |
| 40 | 0.98 | 0.95 | 0.91 | 0.82 | 0.99 | 0.96 | 0.93 | 0.87 |

## F.6 SUBSPACE REGRESSION COMBINED WITH LOBPCG

Subspace regression can be also used to improve results for classical iterative eigensolvers. We demonstrate this for LOBPCG, which is a matrix-free iterative eigensolver that can approximate extremal eigenspaces (Knyazev, 2001). A notable feature of LOBPCG is a possibility of hot start: when approximation to eigenspace of interest is available, it can be used at the initialisation to speed up convergence. In Table 10 we report such speed up for elliptic eigenproblems described in Appendix F.1. Metrics in Table 10are computed for test set, and maximal number of iterations for LOBPCG is set to 1000. Convergence plots are available in FIgure 7a and Figure 7b. It is evident that initialisation by subspace regression lead to both smaller number of iteration and better final error. Note, that the overall cost of method is dominated by the cost of iterations.

Table 10: Performance of LOBPCG with initialisation by subspace regression compared with random initialisation, $N_{\text{it}}$ number of iterations until convergence.

| initialisation | $N_{\text{subspace}}$ | $k_1 = k_2$ | | $k_1 \neq k_2$ | |
|---|---|---|---|---|---|
| | | $N_{\text{it}}$ | relative error $\pm$ std | $N_{\text{it}}$ | relative error $\pm$ std |
| subspace regression | 10 | $410 \pm 245$ | $0.093 \pm 1.109$ | $332 \pm 213$ | $0.028 \pm 0.179$ |
| subspace regression | 20 | $288 \pm 197$ | $0.159 \pm 4.12$ | $249 \pm 174$ | $0.02 \pm 0.087$ |
| subspace regression | 30 | $274 \pm 209$ | $0.047 \pm 0.554$ | $227 \pm 172$ | $0.022 \pm 0.119$ |
| subspace regression | 40 | $263 \pm 200$ | $0.131 \pm 3.218$ | $221 \pm 160$ | $0.02 \pm 0.089$ |
| random | | $685 \pm 265$ | $7.86 \pm 30.16$ | $609 \pm 231$ | $11.42 \pm 35.49$ |

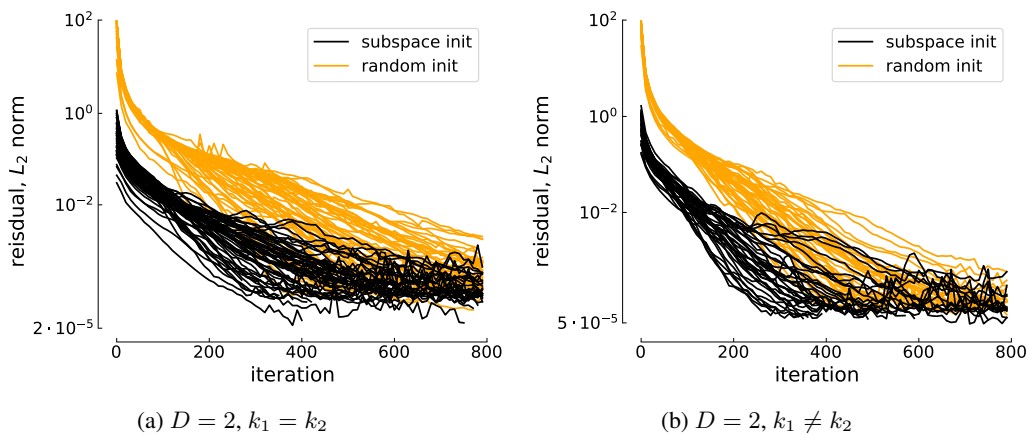

(a) $D = 2$, $k_1 = k_2$

(b) $D = 2$, $k_1 \neq k_2$

Figure 7: Convergence plots for LOBPCG with and without initialisation by subspace regression.

## G    DETAILS ON NUMERICAL EXPERIMENTS FOR PARAMETRIC PDEs

### G.1    DATASETS

For the $D = 2$ stationary diffusion equation we reused datasets described in Appendix F. To generate forcing terms for each $k$ we select 10 eigenvectors $\psi_i, i = 1, \ldots, 10$ corresponding to smallest eigenvalues and compute exact solution as $u = \sum_i \phi_i z_i$ where $z_i \sim N(0, 1)$. Forcing term $f(x)$ corresponding to this solution is $f = \sum_i \frac{1}{\lambda_i} \phi_i z_i$.

For $D = 1 + 1$ Burgers equation (4) we sample random diffusion coefficient $\nu(x)$ and initial condition $u_0(x)$. In both cases we first sample Gaussian random field from $\mathcal{N}\left(0, (\text{id} - \alpha\Delta)^s\right)$. To sample random field $\psi(x)$ for diffusion coefficient we use $\alpha = 40$, $s = 4$. Diffusion coefficient is computed as $\nu = 5 \cdot 10^{-3} + (1 + \tanh(30\psi))/20$ For initial conditions we use random field with $\alpha = 10$, $s = 2$. Uniform grid is used to discretise equation with 128 points for $x$ and 64 points for $t$. Time interval is $\left[0, 10^{-1}\right]$ and $x \in [0, 1]$.

### G.2    ARCHITECTURES AND TRAINING DETAILS

We performed grid search for all methods that involve learning. We start by describing hyperparameters of architectures.

FFNO is used for subspace regression, standard regression, DeepPOD, and for intrusive techniques with basis extraction. Parameters of FFNO used for grid search are described in Appendix F.

DeepONet is used for standard regression, and the intrusive technique with bases extracted from branch net. As a branch net of DeepONet we select a classical convolution network with downsampling by factor of 2 along each dimension and the increase of the number of hidden features by factor of 2. Branch net is defined by the number of features after encoder $N_{e,b}$, kernel size of convolution $k_b$, and number of layers $N_b$. Trunk net is MLP which is defined by the number of hidden neurons $N_{f,t}$, number of layers $N_t$ and the size of basis on the output layer $N_\phi$. In out grid searches we used $N_{e,b} \in [4, 5]$, $k_b \in [3, 7]$, $N_b = 4$, $N_{f,t} = N_\phi \in [100, 200]$, $N_t \in [3, 4]$.

PCANet is defined by the number of POD basis functions used to compress feature and targets $N_{p,f}$ and $N_{p,t}$. Number of MLP layers $N_{MLP}$ and hidden units $N_{p,MLP}$. In our experiments we use $N_{p,f} \in [100, 200, 300, 400, 500]$ for elliptic equation and $N_{p,f} \in [50, 80, 100]$ for Burgers equation, $N_{p,t} \in [100, 200, 300, 400]$ for elliptic equation and $N_{p,t} \in [100, 150, 400]$ for Burgers equation, $N_{MLP} \in [3, 4, 5, 6]$ for elliptic equation and $N_{MLP} \in [3, 5, 7]$ for Burgers equation, $N_{p,MLP} \in [100, 200, 300, 400, 500]$ for elliptic and $N_{p,MLP} \in [100, 300, 500]$ for Burgers equation.

Hyperparameters of kernel methods are the type of kernel and the number of POD basis functions used to compress features and targets $N_{p,f}$ and $N_{p,t}$. We use Matern and RBF kernel and $N_{p,f} \in [50, 100, 150, 200]$, $N_{p,t} \in [50, 100, 150, 200]$

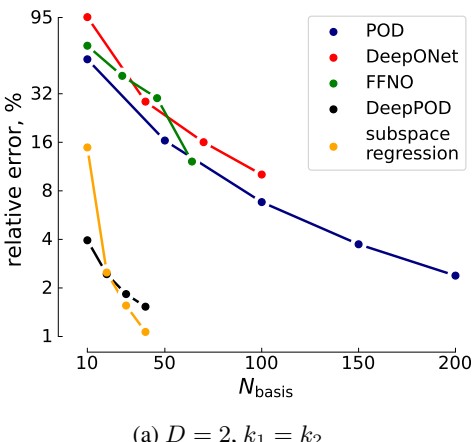
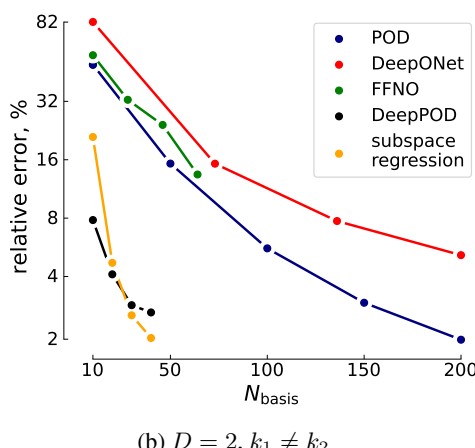

(a) $D = 2, k_1 = k_2$                                (b) $D = 2, k_1 \neq k_2$

Figure 8: Relative errors for two stationary diffusion equations depending on the number of basis functions for several basis construction methods. Bases constructed with DeepPOD and subspace regression lead to the most accurate intrusive methods.

To train neural network we used Lion optimiser with $\text{lr} \in [5 \cdot 10^{-5}, 10^{-4}]$ for FFNO and $\text{lr} \in [10^{-3}, 10^{-4}]$ for all other architectures, $\gamma_{\text{decay}} = 0.5$, $N_{\text{decay}} \in [100, 200]$. We use batch size 10, train PCANet for 3000 epoch and other networks for 1000 epoch.

### G.3 ADDITIONAL RESULTS

We provide two additional results. For elliptic equations we compare optimality of learned bases. The results are in Figure 8a and Figure 8b. Interestingly, global POD leads to a better basis than both FFNO and DeepONet. For DeepONet this is expected, since trunk net does not depend on parameters of PDE. For FFNO the result is more surprising, because the basis is extracted from the last hidden layer, so it explicitly depends on parameters. Bases of DeepPOD and subspace regression are the most optimal one, still they are highly non-optimal when compared with local POD.

For the Burgers equation we compare Deep-POD and two variants of subspace regression in Table 11. SubReg(10) is a subspace regression trained to approximate subspace spanned by first 10 local POD basis functions and SubReg(5) was trained with 5 local POD basis functions. DeepPOD is an unsupervised method and was trained with the whole trajectory. This implies methods are sorted from left to right in the decrease of information they receive about solutions. Interestingly, the SubReg(5) – method, learning the smallest subspace – performs better almost uniformly. A possible explanation is that an optimal subspace of dimension 5 leads to good enough accuracy and is easier to learn than larger subspaces.

Table 11: Relative errors for Burgers equation. Target for SubReg($n$) is subspace of dimension $n$.

| $N_{\text{subspace}}$ | DeepPOD | SubReg(10) | SubReg(5) |
|---|---|---|---|
| 10 | 16.37% | 22.79% | 14.34% |
| 20 | 10.0% | 11.73% | 10.68% |
| 30 | 5.42% | 6.29% | 5.03% |
| 40 | 2.57% | 3.46% | 3.01% |
| 50 | 1.8% | 2.49% | 1.74% |

## H DETAILS ON NUMERICAL EXPERIMENTS FOR ITERATIVE METHODS

### H.1 DEFLATION

Since for a deflation problem one needs to approximate eigenspace spanned by eigenvectors with small eigenvalues, we reused dataset and network trained for elliptic eigenproblem. The description of training and datasets is available in Appendix F. Results in the main text are for $k_1 = k_2$, for $k_1 \neq k_2$ convergence plots are given in Figure 9.

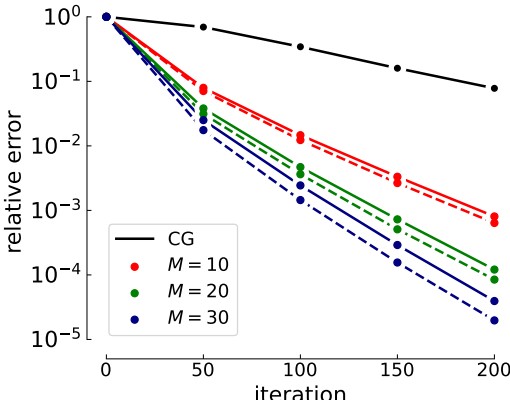

Figure 9: Convergence results for deflated CG, elliptic dataset $D = 2$ with $k_1 \neq k_2$. Learned methods are marked with solid lines, and dashed lines correspond to iterative methods with optimal deflation space, $M$ refers to subspace size.

## H.2 TWO-GRID METHOD

**Elliptic equation dataset**   We consider a 2D elliptic equation on the unit square $\Omega = (0,1)^2$ with homogeneous Dirichlet boundary conditions (5). We aim to learn mapping $k(x) \to \mathcal{S}(V)$. Variability of the dataset comes from the spatially heterogeneous coefficient function $k(x)$.

Each sampled coefficient function is a strictly positive random field built in three steps:

1. Draw i.i.d. Fourier coefficients on a square index set $\mathcal{K} = \{0, \ldots, M-1\}^2$ and form a real field by summing complex exponentials. We additionally introduce a Fourier-space weight $w_k = (1 + \lambda_1 |k|_2^2)^{-1}$ to control the high-frequency components.

2. Multiply the real field from the previous step by $\lambda_2$, then apply a hyperbolic tangent function to control the contrast of the coefficient field values.

3. Rescale the field to the prescribed interval $[\alpha, \beta]$ to ensure strict positivity and enforce a controlled contrast ratio of $\beta/\alpha$.

Exact procedure to generate the 2D field is:

$$s_0(x,y) = \mathrm{Re}\left[ \sum_{k \in \{0, \ldots, M-1\}^2} c_k \frac{e^{\mathrm{i}(k_1 x + k_2 y)}}{1 + \lambda_1 \|k\|_2^2} \right], \quad c_k \sim \mathcal{N}(0,1),$$

$$s(x,y) = \tanh\left( \lambda_2 \cdot s_0(x,y) \right),$$

$$k(x,y) = \alpha + (\beta - \alpha) \frac{s(x,y) + 1}{2}, \quad k(x,y) \in [\alpha, \beta].$$

The equation is descritized on a uniform grid with a 5-point finite-difference stencil, yielding a sparse, symmetric positive-definite matrix. One can observe a sampled normalized coefficient function in Figure 10.

**Target subspace**   We aim to predict a coarse-grid subspace for the two-grid method, which applies a coarse-grid correction

$$x \leftarrow x + V(V^\top A V)^{-1} V^\top (b - Ax)$$

with weighted Jacobi smoothing

$$x \leftarrow x + \omega D^{-1}(b - Ax)$$

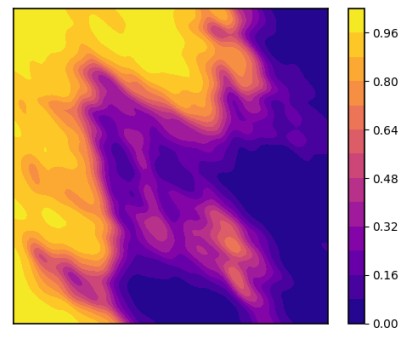

Figure 10: Sample coefficient function.

before and after. This coarse projection $V$ is learned as a problem-specific subspace from the coefficient field. In this setup, our projection matrix $V$ spans the leading eigenspace of the error propagation matrix $I - \omega D^{-1}A$. While the relaxation parameter $\omega$ is used to quickly dampen the fast modes, the coarse-grid projection removes the low-frequency components, resulting in rapid overall convergence. Thus, our target subspace for regression consists of the first few eigenvectors of $I - \omega D^{-1}A$, sorted by the absolute values of their eigenvalues.

**Subspace relaxation**  Initially, we experiment with no under-relaxation (i.e. $\omega = 1.0$) in the Jacobi smoother. As expected, this leads to a target subspace that contains both high- and low-frequency modes. Neural networks are known to have a spectral bias towards low frequencies. In FNO-type models, this spectral bias is especially pronounced because these models truncate high-frequency modes in the Fourier domain. We address this by reducing the relaxation parameter to $\omega = 0.9$, which makes the leading subspace components dominated by slow modes. It is worth noting that this adjustment is consistent with the respective roles of smoothing and coarse-grid correction discussed above. In Figure 11 one can observe the first five eigenvectors of the target subspace for a representative sample under both relaxation settings ($\omega = 1.0$ and $\omega = 0.9$).

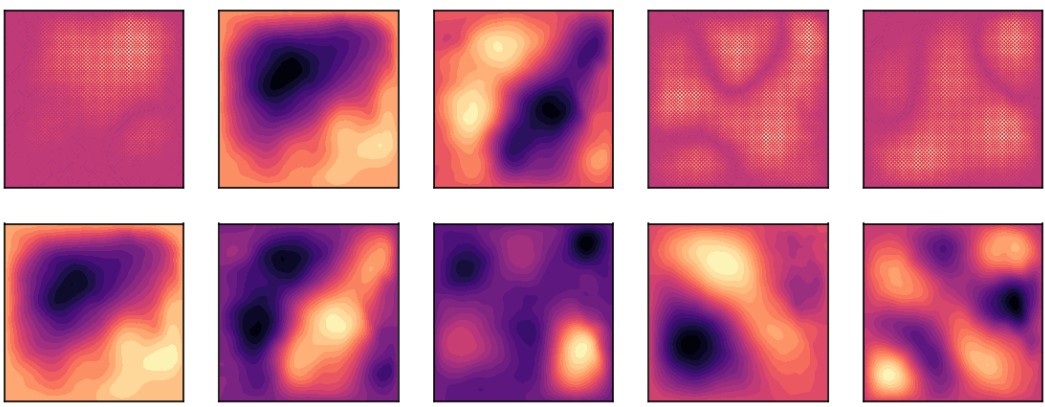

Figure 11: First five eigenvectors of the error propagation matrix $I - \omega D^{-1}A$. Top: $\omega = 1.0$. Bottom: $\omega = 0.9$.

**Subspace prediction**  Before applying the neural network's predicted subspace, we perform a QR decomposition on the predicted matrix $W$ to obtain an orthonormal basis $Q_W$. We assess the quality of the predicted coarse subspace in the two-grid method using three metrics:

1. Cosine angles between the true subspace $V$ and the predicted subspace, computed as the singular values of $Q_W^\top V$. A value closer to 1 indicates better subspace alignment.

2. Relative reconstruction error $e = \min_u \|V - Wu\|_2$ for each true basis vector $V$, computed as $\|(I - Q_W Q_W^\top)V\|_2$. A smaller value indicates that the predicted subspace reconstructs $V_i$ more accurately.

3. Two-grid convergence rate, measured by the spectral radius $\rho$ of the two-grid iteration operator $T$. We estimate $\rho$ via the power method by repeatedly applying $T$ to a vector:

$$v_{k+1} = \frac{Tv_k}{\|Tv_k\|} .$$

A smaller spectral radius indicates faster asymptotic convergence.

In Table 12, we report these metrics for the best-performing models and for the ground-truth target subspace. Across all experiments, the predicted coarse subspaces achieve slightly better two-grid convergence (i.e., lower spectral radius) than the ground-truth subspace. These results are particularly interesting since smaller subspaces yield rather high cosines and reconstruction errors. The increase of size of the predicted subspace improves the quality of the reconstruction and does not degrade in effect on iteration compared to true exact subspace. It is also worth noting that both training objectives yield similarly effective subspaces.

Table 12: Subspace prediction metrics for two-grid method and Jacobi iterations. We report averaged values over the test set. Methods' column values: *Exact subspace* uses the exact leading eigenvectors of error propagation matrix (ground truth). $L_1$ *loss* and $L_2$ *loss* denote subspaces predicted by F-FNO trained with the respective objectives. The Jacobi baseline reports the spectral radius with no coarse correction ($\omega = 1.0$).

| Subspace size | Method | Cosine | Rec. error | Spectral radius |
|---|---|---|---|---|
| – | Jacobi | – | – | 0.9976 |
| 10 | Exact subspace | – | – | 0.9917 |
| | $L_1$ loss | 0.845 | 0.411 | 0.9910 |
| | $L_2$ loss | 0.859 | 0.392 | 0.9908 |
| 20 | Exact subspace | – | – | 0.9858 |
| | $L_1$ loss | 0.960 | 0.222 | 0.9852 |
| | $L_2$ loss | 0.962 | 0.217 | 0.9852 |
| 30 | Exact subspace | – | – | 0.9799 |
| | $L_1$ loss | 0.986 | 0.140 | 0.9790 |
| | $L_2$ loss | 0.986 | 0.139 | 0.9790 |
| 40 | Exact subspace | – | – | 0.9745 |
| | $L_1$ loss | 0.994 | 0.097 | 0.9730 |
| | $L_2$ loss | 0.994 | 0.097 | 0.9731 |

**Data and training details** We generate two different datasets with 32 and 100 interior grid points. Both datasets use $M = 100$ Fourier modes, $\lambda_1 = 0.1$, $\lambda_2 = 1$, interval $[\alpha, \beta] = [1, 50]$, and $\omega = 0.9$ in error propagation matrix. Each dataset contains $1,000$ training and 200 test samples. While a neural network predicts subspace of sizes $\{10, 20, 30, 40\}$, target subspace always contains 10 basis functions.

We train the Factorized Fourier Neural Operator (F-FNO) model (Tran et al., 2021). We first conduct an extensive hyperparameter search on the 32 dataset with:

- Number of retained modes: $\{10, 14, 16\}$.
- Number of processor layers: $\{3, 4, 5\}$.
- Learning rate: $\{10^{-3}, 10^{-4}\}$.
- Step-decay every $\{100, 200\}$ epochs.

By default, the batch size is 64, training runs for $1,000$ epochs, and each processor layer has 64 features. We repeat this search for both $L_1$ and $L_2$ losses and for predicted subspace sizes $\{10, 20, 30, 40\}$. We then select the top-3 hyperparameter configurations per subspace size and loss (by two-grid spectral radius) and train on the dataset with 100 grid points. Throughout the paper, we report results for the best configuration on the dataset with 100 grid points.

# I  SUBSPACE REGRESSION FOR OPTIMAL CONTROL

We consider optimal control of $D = 1 + 1$ heat equation with homogeneous Dirichlet boundary conditions

$$\frac{\partial \phi(x,t)}{\partial t} = \text{div } k \cdot \text{grad } \phi(x,t) - b(x) + \sum_{i=1}^{k} w_i(x) u_i(t),$$

$$y_i = (\psi_i, \phi) \tag{52}$$

$$\min_u L = \frac{1}{2} y(T)^\top y(T) + \frac{\lambda}{2} \int_0^T u(t)^\top u(t).$$

The problem has a simple interpretation. With no control for sufficiently large $T$ system reaches steady-state, which can be computed as a solution of the linear system $\text{div } k \cdot \text{grad } \phi(x,t) = b(x)$. The objective function contains the term $\frac{1}{2} y(T)^\top y(T) \geq 0$. Optimal control minimises amplitude of $y(T)$. Given that $y(t)$ is a projection of state variable $\phi$ on vectors $\psi_i$, the result of optimal control is to reach $\phi$ with $\phi - \sum_i \left( \phi, \widetilde{\psi}_i \right) \widetilde{\psi}_i$ where $\widetilde{\psi}_i$ is any basis in subspace spanned by $\psi_i$.

Control problem (52) is not a linear-quadratic regulator in its standard form since $b(x)$ is present. To get rid of $b(x)$ and use exact solution for linear-quadratic regulator we first discretise PDE (52) and after that introduce additional (constant) variable

$$\frac{d}{dt} \begin{pmatrix} \phi(t) \\ \widetilde{\phi}(t) \end{pmatrix} = \begin{pmatrix} A & -I \\ 0 & 0 \end{pmatrix} \begin{pmatrix} \phi(t) \\ \widetilde{\phi}(t) \end{pmatrix} + \begin{pmatrix} W \\ 0 \end{pmatrix} u, \quad \begin{pmatrix} \phi(0) \\ \widetilde{\phi}(0) \end{pmatrix} = \begin{pmatrix} \phi_0 \\ b(x) \end{pmatrix},$$

$$\min_u L = \frac{1}{2} \phi(T)^\top \Psi \Psi^\top \phi(T) + \frac{\lambda}{2} \int_0^T u(t)^\top u(t). \tag{53}$$

Problem (53) is a standard linear-quadratic regulator, so the exact form of value function is known (Kirk, 2004). Optimal control can be computed as follows

$$\dot{C}_{12} - C_{11} + A C_{12} - \lambda^{-1} C_{11} W W^\top C_{12} = 0, \ C_{12}(T) = 0,$$

$$\dot{C}_{11} + C_{11} A + A C_{11} - \lambda^{-1} C_{11} W W^\top C_{11} = 0, \ C_{11}(T) = \Psi \Psi^\top, \tag{54}$$

$$u(t) = -\lambda^{-1} W^\top \left( C_{11} \phi + C_{12} b(x) \right).$$

As model reduction method we apply balanced truncation closely following (Moore, 2003): (i) solve Lyapunov equations to find controllability and observability Gramians, (ii) find eigendecomposition of controllability Gramian and select coordinates system where controllability Gramian is identity matrix, (iii) find eigendecomposition of observability Gramian and select coordinate system where controllability and observability Gramians coincide, (iv) from the composition of two coordinate transformations build degrees of freedom corresponding to largest eigenvalues of both controllability and observability Gramians.

## I.1  DATASET

We use random gaussian random field $\mathcal{N}\left(0, (\text{id} - \alpha \Delta)^s\right)$ to generate $w_i(x), \psi_i(x)$, diffusion coefficient $k(x)$, initial conditions $u_0(x)$ and forcing $b(x)$. For $w_i, \psi_i, u_0(x), b(x)$ we take $\alpha = 5$ and $n = 4$, $\psi$ and $w_i$ are further orthogonalised with QR, 30 i.i.d. $w_i$ and $\psi_i$ are generated for each dataset sample; for diffusion coefficient we use $\alpha = 6$, $n = 4$ and process generated random field $\chi$ similarly to Burgers equation $k(x) = 5 \times 10^{-3} + (1 + \tanh(5\chi))/10$. Equation is discretised on uniform grid $128 \times 128$, $x \in [0, 1]$, $t \in [0, 5]$. Dataset consists of 1200 samples, 1000 for train, 100 for validation and 100 for test. Optimal reduction by balanced truncation is computed for each sample and later used for subspace regression.

## I.2  ARCHITECTURE AND TRAINING DETAILS

We use FFNO and precisely the same grid search as for the Burgers equation.

Table 13: Results for control.

| $N_{\text{basis}}$ | exact | | $L_1(A, B)$ | | $L_2(A, B; z)$ | |
|---|---|---|---|---|---|---|
| | $E_s$ | $E_o$ | $E_s$ | $E_o$ | $E_s$ | $E_o$ |
| 10 | 4.16% | 4.07% | 12.56% | 12.21% | 10.91% | 10.59% |
| 20 | | | 8.8% | 8.47% | 7.96% | 7.65% |
| 30 | | | 9.49% | 8.94% | 8.74% | 8.42% |
| 40 | | | 7.26% | 6.99% | 7.41% | 7.15% |
| 50 | | | 7.2% | 6.94% | 7.09% | 6.88% |

### I.3 RESULTS

Results are summarised in Table 13. We train a neural network with two subspace regression losses on first 10 basis vectors obtained with balanced truncation. As metrics we use relative observation error $E_o$ at time $T$ and relative full state error $E_s$ at time $T$. One can observe that subspace embedding techniques improve accuracy for both loss functions. Interestingly, $L_2$ leads to slightly better error for small subspace sizes. Overall accuracy is acceptable but does not reach optimal performance reported in the first columns.

## J SENSITIVITY TO TRAIN-TEST SPLIT AND THE CHOICE OF HYPERPARAMETERS

All results in the main text are reported without error bars, that are usually computed by varying random initialisation or train-test split. All our experiments involve extensive hyperparameter search, so computing statistics for distinct initialisations or train-test split would require order of magnitude more compute. Here we demonstrate for elliptic eigenproblem (Appendix F.1) and Burgers equation (Appendix G.1) that sensetivity to train-test split is relatively small and not significant to main conclusions reached by analysis of "single-run" results.

Table 14: Sensitivity to train-test split for elliptic eigenproblem and Burgers equation, subspace regression trained with $L_2(A, B; z)$ loss function.

| $N_{\text{subspace}}$ | elliptic, $k_1 = k_2$ | | Burgers | |
|---|---|---|---|---|
| | train error ± std, % | test error ± std, % | train error ± std, % | test error ± std, % |
| 10 | $24.37 \pm 1.47$ | $30.38 \pm 1.34$ | $25.5 \pm 1.17$ | $25.71 \pm 1.72$ |
| 20 | $5.69 \pm 0.53$ | $6.46 \pm 0.26$ | $11.15 \pm 1.2$ | $11.0 \pm 1.43$ |
| 30 | $3.34 \pm 0.26$ | $3.79 \pm 0.35$ | $7.04 \pm 0.35$ | $7.01 \pm 0.64$ |
| 40 | $1.84 \pm 0.12$ | $2.12 \pm 0.11$ | $4.56 \pm 0.32$ | $4.72 \pm 0.61$ |

Table 15: Sensitivity to train-test split for elliptic eigenproblem and Burgers equation, subspace regression trained with $L_2(A, B; z)$ loss function.

| $N_{\text{subspace}}$ | elliptic, $k_1 = k_2$ | | Burgers | |
|---|---|---|---|---|
| | train error [min, max], % | test error [min, max], % | train error [min, max], % | test error [min, max], % |
| 10 | $[23.81, 37.8]$ | $[28.67, 39.05]$ | $[22.01, 32.64]$ | $[23.79, 34.62]$ |
| 20 | $[4.74, 100.0]$ | $[5.51, 100.0]$ | $[9.53, 20.14]$ | $[10.06, 21.05]$ |
| 30 | $[2.77, 8.6]$ | $[3.33, 9.06]$ | $[5.58, 21.81]$ | $[5.84, 21.7]$ |
| 40 | $[2.06, 7.05]$ | $[2.31, 7.34]$ | $[4.3, 47.43]$ | $[4.31, 283.33]$ |

For both Burgers and elliptic equations we randomly split dataset on train and test set 5 times and report mean metrics and standard deviation. Networks are trained for hyperparameters found by grid search. The results are available in Table 14. Variability is clearly present, but it is not pronounced enough.

Variability to train-test split should be compared to sensitivity to the selection of hyperparameters reported in Table 15. We see that for certain cases the span of best and worse performance reaches $100\%$ which is much higher than variability to train-test split. Note that for elliptic equation grid search was performed on downsampled dataset with $32 \times 32$ grid.

