# OpenReview forum: "Deep Learning for Subspace Regression"
_ICLR.cc/2026/Conference — ICLR 2026 Poster_

### Official Review · Reviewer_jxox · 2025-10-27

**Soundness:** 3
**Presentation:** 3
**Contribution:** 3
**Rating:** 8
**Confidence:** 3

**Summary:**

The problem and a method of subspace regression (using deep learning) are proposed. The problem is to predict a subspace (i.e., an element of a Grassmann manifold) from some inputs. The authors propose loss functions, use cases of neural net-based architectures, and a way to facilitate training by regressing subspaces with larger dimensionality. The numerical examples also support the utility of the proposed framework.

**Strengths:**

The motivation is clearly explained with concrete examples, the proposed framework is technically sound, and the numerical examples are convincing to show the approach's effectiveness.

I assume the problem setting itself is something new, but as I am not an expert in the area, I cannot fully assess the novelty in the particular context of subspace regression.

**Weaknesses:**

I basically don't see major issues.

Although the paper is carefully written, a kind of schematic drawing of (an instance of) the problem and the method will be highly appreciated to facilitate smoother understanding.

**Questions:**

What are the most relevant studies, particularly in terms of the problem setting, i.e., regression to subspaces? As the paper does not have a distinct Related Work section, it is a little difficult to follow the exact technical context.

---

> ### Author Response · Authors · 2025-11-20
> **part 1**
>
> We thank the reviewer for reading the manuscript and providing several suggestions.
>
> > Although the paper is carefully written, a kind of schematic drawing of (an instance of) the problem and the method will be highly appreciated to facilitate smoother understanding.
>
> We agree that it would be great to have a simplified schematic description of the approach. Unfortunately, the page limit required by the rules of ICLR does not allow us to introduce such a scheme in the main text.
>
> In principle, our approach is a straightforward regression problem with a mildly unusual target. The readers may assume that we have a neural network that predict a tall matrix from some data specific to the problem (e.g., diffusion coefficient for elliptic eigenproblem).
>
> > What are the most relevant studies, particularly in terms of the problem setting, i.e., regression to subspaces? As the paper does not have a distinct Related Work section, it is a little difficult to follow the exact technical context.
>
> We agree that we do not have a dedicated Related Work section. In place of that we discussed relevant works throughout the paper. Below we provide a discussion which we can, if the reviewer insists, add to the appendix of the paper.
>
> In the paper we introduce regression on grassmannian. The most related approach is interpolation [2]. Interpolation is usually defined on tangent space: (i) logarithm map is applied to build local coordinate chart, (ii) standard interpolation technique is used in the tangent space, (iii) exponential map is applied to transfer interpolated element of tangent space to the point on grassmannian [3]. This general form of manifold interpolation is particularly convenient to apply to Grassmann manifolds since both logarithm and exponential maps are available in closed form [2]. Higher order interpolation methods including Hermite interpolation can be defined in the same way[4].
>
> A standard application of interpolation on grassmannian is MOR [3], [4], especially POD [8]. For MOR applications, it is often advantageous to perform clustering in the space of parameters [6], and use more advanced statistical models for interpolation [5].
>
> Interpolation implies the basis is computed based on classical methods. An alternative is to learn the basis directly from data [7], [8]. In [7] the basis is extracted from DeepONet architecture and in [8] a basis is learned for stationary problems by optimising an orthogonal projector.
>
> [1] -- Deep orthogonal decomposition: a continuously adaptive data-driven approach to model order reduction, https://arxiv.org/abs/2404.18841
>
> [2] -- A gentle introduction to interpolation on the Grassmann manifold, https://www.mate.polimi.it/biblioteca/add/qmox/03-2024.pdf
>
> [3] -- Manifold interpolation and model reduction, https://arxiv.org/abs/1902.06502
>
> [4] -- Interpolation on manifolds of CFD-based fluid and finite element-based structural reduced-order models for on-line aeroelastic predictions, https://stacks.stanford.edu/file/druid:dw083rz0825/amsallem_thesis-augmented.pdf
>
> [5] -- Data-driven surrogates for high dimensional models using Gaussian process regression on the Grassmann manifold, https://arxiv.org/abs/2003.11910
>
> [6] -- Nonlinear model order reduction based on local reduced-order bases, https://onlinelibrary.wiley.com/doi/abs/10.1002/nme.4371
>
> [7] -- Machine-learning custom-made basis functions for partial differential equations, https://arxiv.org/abs/2111.05307
>
> [8] -- Proper orthogonal decomposition: Theory and reduced-order modelling, Stefan Volkwein.

---

### Official Review · Reviewer_MfjB · 2025-10-29

**Soundness:** 3
**Presentation:** 3
**Contribution:** 4
**Rating:** 8
**Confidence:** 3

**Summary:**

This work proposes a reliable approach for subspace regression in ROM problems. To address the difficulty of recovering the exact number of subspace bases, the authors introduce a relaxation named subspace embedding that learns a higher-dimensional subspace instead, intended to sufficiently encompass the target lower-dimensional subspace and improve both accuracy and robustness at the expense of some gap between the dimensions of optimal and learned subspaces. This new formulation is evaluated in several applications and compared to different baselines to highlight its pros and cons.

**Strengths:**

1. Well-structured paper with clear problem definition, rigorous theoretical analysis, thorough experiments and meticulous evaluation.
2. The subspace embedding method is novel and seems easily generalizable to different ROM problems with good robustness.
3. Different implementations of the formulation are given in Theorem 1.
3. Both pros and cons of this method are discussed in detail, the explanations are insightful and intuitive.
4. Several nice illustrations helping readers appreciate the theoretical analysis in the appendix.

**Weaknesses:**

Overall, this is a solid paper without significant flaws. Just a few minor ones. I'm willing to increase the score if they are addressed and my questions are answered.

1. Seems like a $\psi(x)$ next to $U(x)$ is missing in (3).
2. I think a clear definition of "relative error" is needed. See Question 1 below.
3. Use \citep and \citet properly.
4. Some sentences could be streamlined and a few typos can be fixed.

**Questions:**

1. How is "relative error" defined? $W_\theta$ and $V$ should have different ranks ($r$ and $k$), right?
   * Maybe related to the Frobenius norm in Theorem 2?
   * Is the definition of "relative error" the same for different models in Figure 2?
2. Will subspace embedding perform worse as the ambient dimension $n$ increases? I assume that as $n$ increases, the low-dim linear subspaces will be more likely orthogonal with one another and make subspace embedding trickier. Is this intuition reasonable?
   * In other words, what would Figure 1 approximately look like if $n$ is the third dimension apart from $r$ and error?
3. How impactful is the gap $r-k$ to different ROM problems when the predicted larger subspace is used?
   * Any postprocess for trimming the predicted subspace to reduce its rank to $k$?
4. How trainable or reusable is a subspace regressor for a collection of problems of varying dimension $n$ or domain $\Omega$?
   * For instance, FEA is widely used to solve PDEs over domains of complicated geometries, and its approximate solution often takes the form on Line 107. How applicable is your method considering that the number of $\phi_i$ often changes due to either the varying nonlinearity in the solution field or the varying $\Omega$?

---

> ### Author Response · Authors · 2025-11-20
> **part 1**
>
> We would like to thank the reviewer for many useful suggestions. Below we explain changes we introduce in the text based on te
>
> > Seems like a $\psi(x)$ next to $U(x)$ is missing in (3).
>
> The wavefunction was missing, it is added in the revised version of the manuscript.
>
> > I think a clear definition of "relative error" is needed. See Question 1 below.
>
> We agree that in the previous version of the manuscript it was unclear what relative error we report. This is now explained in the beginning of the section with numerical experiments.
>
> > Use \citep and \citet properly.
>
> All of our citations are parenthetical, so. we swapped \cite to citep
>
> > How is "relative error" defined? $W_\theta$ and $V$ should have different ranks ($r$ and $k$), right?
> > Maybe related to the Frobenius norm in Theorem 2?
> > Is the definition of "relative error" the same for different models in Figure 2?
>
> We agree that it is important to clarify what we mean by relative error. We insert the following explanation in the beginning of Section 4:
> "For most problems we report relative error measured in percents. For predicted quantity $\widetilde{v}$ and ground truth value $v$ it reads $100\\%\times\left\\|v - \widetilde{v}\right\\|_2 \big/\left\\| v\right\\|_2$. For eigenvalue problems the numerator is $\mathbb{Z}_2$ adjusted as explained below. For iterative methods we report convergence plots for relative error, without additional factor $100\\%$."
>
> So, for all problems the same error is reported which is related not to the subspaces directly but rather to the downstream problem. For example, for eigenproblem the procedure is as follows:
> 1. Predict subspace $W$
> 2. Solve reduced eigenproblem problem $W^\top A W z = \lambda z$
> 3. Restore approximate solution in the original space ${x}^{'}\_{0} = W z\_{0}$ (eigenvector with the smallest eigenvalue).
> 4. Compute relative error $\min_{\pm}\left\\|x_0 \pm {x}^{'}_0\right\\| \big/ \left\\|x_0\right\\|$.

---

> ### Author Response · Authors · 2025-11-20
> **part 2**
>
> > Will subspace embedding perform worse as the ambient dimension $n$ increases? I assume that as $n$ increases, the low-dim linear subspaces will be more likely orthogonal with one another and make subspace embedding trickier. Is this intuition reasonable?
> >
> > In other words, what would Figure 1 approximately look like if $n$ is the third dimension apart from $r$ and error?
>
> We thank the reviewer for an interesting question.
>
> First, we would like to note that we are generally working with high-dimensional spaces. For eigenvalue problems $n = 10^{4}$ and random linear subspaces in this space are likely to be orthogonal.
>
> If we measure mean angles between spaces for elliptic eigenproblem we observe
>
> | dataset                            | mean cos $\pm$ std |
> | ---------------------------------- | ------------------ |
> | $k_1 = k_2$, $n = 32\times32$      | $0.5\pm0.18$       |
> | $k_1 \neq k_2$, $n = 32\times32$   | $0.52\pm0.12$      |
> | $k_1 = k_2$, $n = 100\times100$    | $0.51\pm0.18$      |
> | $k_1 \neq k_2$, $n = 100\times100$ | $0.53\pm0.12$      |
>
> Where the result for $n=32\times 32$ is for a dataset generated from the same distribution but on the smaller grid.
>
> As one can see the angles are about $58\degree$ far from orthogonal. This is because we consider very specific subspaces that are related to each other, not random subspaces.
>
> If we evaluate subspace embedding technique on these two problems we find:
>
> $100\times100$, $k_1 = k_2$
>
> | $N_{\text{subspace}}$ | best train error, $\\%$ | best test error, $\\%$ |
> | --------------------- | ---------------------- | --------------------- |
> | $10$                  | $30.53$                | $40.72$               |
> | $20$                  | $6.6$                  | $8.98$                |
> | $30$                  | $4.96$                 | $6.28$                |
> | $40$                  | $3.55$                 | $4.67$                |
>
> $32\times32$,  $k_1 = k_2$
>
> | $N_{\text{subspace}}$ | best train error, $\\%$ | best test error, $\\%$ |
> | --------------------- | ---------------------- | --------------------- |
> | $10$                  | $29.91$                | $39.18$               |
> | $20$                  | $6.72$                 | $9.24$                |
> | $30$                  | $3.99$                 | $6.58$                |
> | $40$                  | $3.48$                 | $5.45$                |
>
> So, embedding works roughly the same, regardless of resolution.
>
> The problem is $n$ itself is not directly related to the complexity of the dataset.
>
> For example, we can generate less diverse diffusion coefficient on the grid $100\times 100$ and this gives us
>
> $n =100\times100$, $k_1 = k_2$, simplified dataset
>
> | $N_{\text{subspace}}$ | best train error, $\\%$ | best test error, $\\%$ |
> | --------------------- | ---------------------- | --------------------- |
> | $10$                  | $5.21$                 | $5.89$                |
> | $20$                  | $0.24$                 | $0.25$                |
> | $30$                  | $0.13$                 | $0.14$                |
> | $40$                  | $0.09$                 | $0.09$                |
>
> $n =100\times100$, $k_1 \neq k_2$, dataset from the paper
>
> | $N_{\text{subspace}}$ | best train error, $\\%$ | best test error, $\\%$ |
> | --------------------- | ---------------------- | --------------------- |
> | $10$                  | $32.49$                | $39.28$               |
> | $20$                  | $8.85$                 | $10.8$                |
> | $30$                  | $5.9$                  | $6.89$                |
> | $40$                  | $5.29$                 | $6.09$                |
>
> We can make several observations:
> 1. $n$ is the same but performance is quite different
> 2. Subspace embedding works well for both datasets
> 3. Subspace embedding works much better for the simplified dataset
>
> So, to answer your last question, relative errors on Figure 1 are not going to change with the increase of $n$ for the same dataset. The reason is FFNO has spectral convolution in it which provides roughly the same accuracy regardless of resolution.

---

> ### Author Response · Authors · 2025-11-20
> **part 3**
>
> > How impactful is the gap $r-k$ to different ROM problems when the predicted larger subspace is used?
> > Any postprocess for trimming the predicted subspace to reduce its rank to $k$?
>
> This is also an interesting question with a rather subtle answer. In general we expect accuracy to increase when larger subspaces are used. We also observe an interesting pattern: sometimes it is possible to increase accuracy by decreasing the dimension of subspace in training data. An example appear in Table 11 (in the revision) in Appendix G. A part of this table is reproduced below
>
> | $N_{\text{subspace}}$ | SubReg$(10)$ | SubReg$(5)$ |
> | --------------------- | ------------ | ----------- |
> | $10$                  | $22.79\%$    | $14.34\%$   |
> | $20$                  | $11.73\%$    | $10.68\%$   |
> | $30$                  | $6.29\%$     | $5.03\%$    |
> | $40$                  | $3.46\%$     | $3.01\%$    |
> | $50$                  | $2.49\%$     | $1.74\%$    |
>
> SubReg$(10)$ and SubReg$(5)$ are similar architectures trained with the same loss function but on $10$ and $5$ ground truth vectors computed with POD. We see that the model with access to smaller subspace somewhat counterintuitively performs better.
>
> The example is also related to the second question: is it possible to trim subspaces? The answer is negative. In the scheme of the subspace regression subspaces are treated as primal objects, with no possibility to assign weights to a particular direction. In the example above improved accuracy is achieved by trimming the data in the pre-processing stage. This implies SubReg$(10)$ spend resources trying to predict $10$ basis vectors that have the same weight in the loss function. However, the contribution of these $10$ vectors to the error of approximation is not uniform. By construction first $5$ vectors have a dominant contribution.
>
> > How trainable or reusable is a subspace regressor for a collection of problems of varying dimension $n$ or domain $\Omega$?
> >
> > For instance, FEA is widely used to solve PDEs over domains of complicated geometries, and its approximate solution often takes the form on Line 107. How applicable is your method considering that the number of $\phi_i$ often changes due to either the varying nonlinearity in the solution field or the varying $\Omega$?
>
> We believe our method is applicable to complex geometries since operator learning has being shown to work in such cases [1], [2]. Additional experiments for complex geometries will require substantial change in the architecture, so we are unable to present additional results at this time.
>
> As for reusability for different $n$, we perform additional experiments that confirm the possibility to train on small $n$ and apply a model on larger $n$. It appears in Appendix F.4 (Table 6) in the revised manuscript. The test was performed for a QM problem in $D=1$. The network was trained on grid $N_x = 100$ and evaluated on grids with resolution up to $N_x=500$.
>
> | $N_x$ | test error, $\\%$ |
> | ----- | ---------------- |
> | $100$ | $0.83$           |
> | $150$ | $1.37$           |
> | $200$ | $2.12$           |
> | $250$ | $2.95$           |
> | $300$ | $3.96$           |
> | $350$ | $4.32$           |
> | $400$ | $5.33$           |
> | $450$ | $5.32$           |
> | $500$ | $6.79$           |
>
> As we see, the results are reusable, but some degradation in relative accuracy is observed.
>
> [1] -- Geometry-Informed Neural Operator for Large-Scale 3D PDEs, https://arxiv.org/abs/2309.00583
> [2] -- GNOT: A General Neural Operator Transformer for Operator Learning, https://arxiv.org/abs/2302.14376

---

> > ### Comment · Reviewer_MfjB · 2025-11-21
> >
> > I appreciate the authors' detailed answer supported by experimental evidence. Thereby I increase my score.

---

### Official Review · Reviewer_fKUi · 2025-10-30

**Soundness:** 2
**Presentation:** 1
**Contribution:** 2
**Rating:** 4
**Confidence:** 3

**Summary:**

The paper studies a subspace regression problem that aims to learn a mapping from a parameter set to a manifold of subspaces. One central challenge is that the target of the regression is itself a *set of spaces*, rather than points in a Euclidean domain. The main contributions lie in designing appropriate loss functions and embedding techniques that allow these spaces to be compared, optimized over, and represented effectively within a learning pipeline.

**Strengths:**

The problem is challenging, and the numerical results appear promising.

**Weaknesses:**

There is no comparison with classical non–learning methods. For example, in the eigenvalue problem, when approximating the smallest eigenvalue, conventional numerical approaches are already highly efficient and accurate. Without such baselines, it is difficult to assess whether the proposed learning framework offers any practical advantage.

The presentation is weak, making the article difficult to read. Beyond apparent typos (see below), the referee identifies the following concerns:

   1. Above Eq. (1), both matrices and parametric models are denoted by $W$, which is confusing.

   2. Above Eq. (1), in the general setting, it would help to specify $V(r)$ explicitly for each example discussed later.

   3. In the approximate eigenspaces example, $U$ and $E$ should be defined. On line 113, the object described is not a subspace.

   4. On line 141, $V$ should be a matrix rather than a subspace based on context. If $V$ were a subspace, $S(V)$ would be ill-defined.

   5. In Theorem 3, the mappings $F$ and $G$ should be indexed as $F_k$ and $G_k$, since they are $k$-dependent. The quantities $\#_F(k, D)$ and $\#_G(k, D)$ are not defined.

   6. The sentence “As the main neural network architecture we used FFNO Tran et al. (2021), a descendant of FNO Li et al. (2020), and performed extensive grid searches for all experiments” is unclear; its relevance and meaning should be clarified.


Numerous grammar issues:

   1. Line 107: “to approximates” → “to approximate”
   2. Line 112: “is solver repeatedly” → unclear phrasing
   3. Line 206: “we allowed” → revise for tense consistency
   4. Line 261: “Theorem 3 suggest” → “Theorem 3 suggests”
   5. Line 304: missing punctuation

   …and many others throughout the manuscript.

**Questions:**

There are several technically unclear points in the presentation:

1. Line 209: the meaning of “interpolation” is unclear and should be defined precisely in this context.

2. Theorem 3: it is not explained why the functions $F_k$ and $G_k$ are piecewise constant, nor why they take distinct values. A justification or reference is needed.

3. The description of the **SUBSPACE EMBEDDING** technique is very unclear. How does it work in practice? Does it imply that the learnable data consist of higher-dimensional subspaces, and if so, how are they embedded and compared?

4. In the tables of numerical results, the meaning of the reported percentages is not explained. Do they represent accuracy, relative error, or improvement over a baseline?

---

> ### Author Response · Authors · 2025-11-20
> **part 1**
>
> We thank the reviewer for providing constructive critique of our contribution. Below we describe changes we incorporate into the revised manuscript to address reviewer's suggestions.
>
> >  For example, in the eigenvalue problem, when approximating the smallest eigenvalue, conventional numerical approaches are already highly efficient and accurate. Without such baselines, it is difficult to assess whether the proposed learning framework offers any practical advantage.
>
> We include additional non-learning baselines for eigenproblems. A highly efficient and versatile method for the approximation of extremal eigenspace is LOBPCG [1]. A notable feature of the method is the possibility of "hot start". That is, if approximation to the eigenspace is known, LOBPCG can use it as initial guess and converge faster.
>
> We perform two types of experiments with LOBPCG. For all  experiments LOBPCG allows to make at most $1000$ iteration. In the first type of experiments a solution predicted by subspace regression is used to initialize LOBPCG, in the second type of experiments standard random initialisation is used. Below average (with respect to test set) relative errors $\pm$ standard deviations are reported for two elliptic eigenproblems.
>
> $k_1 \neq k_2$
>
> | init                | $N_{\text{subspace}}$ | iterations to converge | relative error $\pm$ std, $\\%$ |
> | ------------------- | --------------------- | ---------------------- | ------------------------------ |
> | subspace regression | $10$                  | $332\pm213$            | $0.028\pm0.179$                |
> | subspace regression | $20$                  | $249\pm174$            | $0.02\pm0.087$                 |
> | subspace regression | $30$                  | $227\pm172$            | $0.022\pm0.119$                |
> | subspace regression | $40$                  | $221\pm160$            | $0.02\pm0.089$                 |
> | random              |                       | $609\pm231$            | $11.42\pm35.49$                |
>
> $k_1 = k_2$
>
> | init                | $N_{\text{subspace}}$ | iterations to converge | relative error $\pm$ std, $\\%$ |
> | ------------------- | --------------------- | ---------------------- | ------------------------------ |
> | subspace regression | $10$                  | $410\pm245$            | $0.093\pm1.109$                |
> | subspace regression | $20$                  | $288\pm197$            | $0.159\pm4.12$                 |
> | subspace regression | $30$                  | $274\pm209$            | $0.047\pm0.554$                |
> | subspace regression | $40$                  | $263\pm200$            | $0.131\pm3.218$                |
> | random              |                       | $685\pm265$            | $7.86\pm30.16$                 |
>
> For both eigenproblems initialisation by subspace regression lead to significantly better error and faster convergence. Note, that subspace regression is only used on the initialisation stage, with negligible numerical cost (on the order of $\sim 10$ LOBPCG iterations).
>
> In the revised manuscript these results are reported in Appendix F.5. and in Section 4.1.
>
> [1] -- Toward the Optimal Preconditioned Eigensolver: Locally Optimal Block Preconditioned Conjugate Gradient Method, Andrew V. Knyazev, https://epubs.siam.org/doi/10.1137/S1064827500366124.
>
> > There is no comparison with classical non–learning methods.
>
> We do not find this claim of the reviewer to be completely fair. A few classical non-learning methods were used as benchmarks:
> 1. We compare the performance of subspace regression with classical interpolation on Grassmann manifolds (Table 1).
> 2. To build reduced models we use classical POD which is a classical non-learning technique (Figure 2).
> 3. Similarly the kernel method, a non-parametric approach, is on the list of baselines (Figure 2).
> 4. For iterative methods we compare against oracle that has information about larger subspace ($20$, $30$ for deflation and $20$, $30$, $40$ for relaxation) and found that subspace regression is competitive in this setting even though the training data includes only subspace of dimension $10$.

---

> ### Author Response · Authors · 2025-11-20
> **part 2**
>
> We would like to thank the reviewer for finding several typos and inaccuracies in the presentation. We introduce the following changes in response:
>
> > Above Eq. (1), both matrices and parametric models are denoted by $W$, which is confusing.
>
> We change the name of the parametric model to $Y_{\theta}$.
>
> > Above Eq. (1), in the general setting, it would help to specify $V(r)$ explicitly for each example discussed later.
>
> We find it difficult to incorporate this suggestion.
>
> First, above equation (1) the reader does not know much about the examples discussed later. These examples are only mentioned and to specify $V(r)$ we will need to first introduce all problems we are dealing with: matrices, operators, PDEs, etc.
>
> Second, $V(r)$ is already described for each example. The general structure of each description in Section 2.2. is: (i) the description and motivation of the problem, (ii) the description of the mapping we need to approximate. We provide several excerpts from these sections:
> 1. (Intrusive POD for time-dependent PDEs) "We apply subspace regression with POD to learn the function that maps PDE data to the subspace formed by reduced basis $\left\\{\psi_1,\dots,\psi_{k}\right\\}$ for some small $k$."
> 2. (Coarse grid correction for iterative methods) "Naturally, subspace regression (1) for this problem approximates the mapping $A\rightarrow \mathcal{S}(V)$ or $k(x) \rightarrow \mathcal{S}(V)$ for linear systems resulting from equation (5)."
> 3. (Deflation for conjugate gradient) "In analogy with previous example, the subspace regression problem considered approximates $A\rightarrow \mathcal{S}(V)$ or $k(x) \rightarrow \mathcal{S}(V)$, but this time $V$ spans eigenspaces with small eigenvalues of matrix $A$."
> 4. (Balanced-truncation for linear-quadratic control) "In this case the goal of subspace regression~(\ref{eq:subspace regression}) is to approximate the mapping $A, B, C \rightarrow \mathcal{S}\left(\overline{\mathcal{T}}\right)$, where $\overline{\mathcal{T}}$ is tall matrix assembled from first few columns of $\mathcal{T}$."
> An example on eigenspace prediction will be discussed below, since reviewers raised separate concerns about it.
>
> As one can see, each example explicitly discusses the mapping $V(r)$ that is approximated in the subspace regression problem.
>
> > In the approximate eigenspaces example, $U$ and $E$ should be defined.
>
> We agree that the knowledge about Schrödinger equation depends on the background of the reader, so we provide additional clarification that $U(x)$ is potential energy and $E$ is the energy of the system (eigenvalue of Schrödinger operator).
>
> > On line 113, the object described is not a subspace.
>
> Initially, the object on line 113 was defined as $U(x) \rightarrow \left\\{\sum_{i=1}^{K}\alpha_i\phi_{i}(x) = f, \alpha_i \in \mathbb{R}\right\\}$. We assumed that the object on the right is a set that contain all functions of the form $\sum_{i=1}^{K}\alpha_i\phi_{i}(x) = f$, which mean it is a subspace of dimension $K$ (given that eigenfunctions of self-adjoint operator are orthogonal) and $\phi_i, i=1,\dots,K$ is a basis of this subspace.
>
> To improve presentation we change it to $U(x) \rightarrow \left\\{f(x): f(x)=\sum_{i=1}^{K}\alpha_i\phi_i(x), \alpha_i \in \mathbb{C}\right\\}$ and add in words that "That is, we wish to predict subspace spanned by first $K$ eigevectors".
>
> > On line 141, $V$ should be a matrix rather than a subspace based on context. If $V$ were a subspace, $S(V)$ would be ill-defined.
>
> We agree that this part may sound confusing, but actually the problem has right $GL$ invariance as we show below. This fact is used often in numerical linear algebra, MOR literature and numerical solution of PDEs (weak formulation).
>
> If we have linear equation $Ax = b$ with symmetric positive definite matrix $A$ and try to find solution in the form $x = V z$, where $V$ is tall full rank matrix, we enforce Petrov-Galerkin condition to find $z$, that is $V^{\top}\left(b - A V z\right) = 0$. In other words, the residual is orthogonal to the range of $V$. In this case the solution reads $z = \left(V^\top A V\right)^{-1} V^\top b$ and for $x$ we find $x = V\left(V^\top A V\right)^{-1} V^\top b$. Now, suppose we change the basis in the subspace spanned by columns of $V$. This is equivalent to multiplying $V$ from the right by an invertible square matrix $G$. If we plug in a new matrix $W = V G$ into the solution above we find
> $$
> W\left(W^\top A W\right)^{-1} W^\top b = VG\left((VG)^\top A VG\right)^{-1} (VG)^\top b = V\left(V^\top A V\right)^{-1} V^\top b.
> $$
> This means that only a subspace formed by matrix $V$ matters, not a particular basis that was used to form columns of $V$.
>
> We add that we mean subspace formed by columns of matrix $V$.
>
> > In Theorem 3, the mappings $F$ and $G$ should be indexed as $F_k$ and $G_k$, since they are $k$-dependent.
>
> We add subscript $k$ as the reviewer suggested.

---

> ### Author Response · Authors · 2025-11-20
> **part 3**
>
> > The quantities $\\#_F(k, D)$ and $\\#_G(k, D)$ are not defined
>
> We can not agree with this statement. In the previous version of the manuscript  $\\#_F(k, D)$ and $\\#_G(k, D)$ was defined in the first statement of Theorem 1:
> 1. $F, G$ are piecewise constant functions that take $\\#\_{F}(k, D),\\#\_{G}(k, D) < \infty$ distinct values.
>
> So it is said that $F, G$ are piecewise constant functions and $\\#\_{F}(k, D),\\#\_{G}(k, D)$ are the number of distinct values these piecewise constant functions may have. In other words $F$ and $G$ map each real number to some element of finite set and $\\#_{F}(k, D),\\#\_{G}(k, D)$ are the number of elements in these sets.
>
> We now change this statement 1. to the following more accurate version:
> 1. $F_{k}, G_{k}$ are piecewise constant functions that map real numbers to elements of sets $S_{F_{k}}, S_{G_{k}}$. Sets $S_{F_{k}}, S_{G_{k}}$ are finite with $\\#\_{F\_{k}}(k, D), \\#\_{G\_{k}}(k, D)$ distinct elements.
>
> >The sentence “As the main neural (...)” is unclear; its relevance and meaning should be clarified
>
> We agree that this phrase may be unclear. The intended meaning of this phrase was:
> 1. We use FFNO architecture that is a modified version of FNO
> 2. We performed grid search
>
> We rephrase this as "In all experiments we used FFNO architecture, a modification of FNO, and performed extensive hyperparameter grid search."
>
> We appreciate that reviewer identified several grammar issues
>
> >Line 107: “to approximates” → “to approximate”
>
> >Line 112: “is solver repeatedly” → unclear phrasing
>
> >Line 206: “we allowed” → revise for tense consistency
>
> >Line 261: “Theorem 3 suggest” → “Theorem 3 suggests”
>
> >Line 304: missing punctuation
>
> We change these phrases according to the reviewers suggestions. The phrase “is solver repeatedly” is changed to "is solved repeatedly".
>
> > Line 209: the meaning of “interpolation” is unclear and should be defined precisely in this context.
>
> Phrase "after interpolation subspace is used to construct a reduced model" appears in the following context:
> "It is justified because of two unique properties of regression and interpolation on grassmanian: (i) inclusion of vector subspaces is well-defined; (ii) after interpolation subspace is used to construct a reduced model."
> So it refers to "interpolation on grassmanian" which is a well defined concept. For example, we have a function $V: \mathbb{R} \rightarrow \text{Gr}(r, n)$. We know the values of this function at several points $V(x_1), \dots, V(x_{N})$. By interpolation one means a numerical procedure that find $V(x)$ at arbitrary point between $\min\left\\{x_1, \dots, x_N\right\\}$ and $\max\left\\{x_1, \dots, x_N\right\\}$ using available observations $V(x_1), \dots, V(x_{N})$.
>
> So the meaning of interpolation is standard.
>
> We slightly rephrase this fragment as:
> "It is justified because of two unique properties of regression and interpolation on grassmanian: (i) inclusion of vector subspaces is well-defined; (ii) subspace, predicted by regression model or interpolated by standard techniques, is used to construct a reduced model."
>
> > Theorem 3: it is not explained why the functions $F_k$ and $G_k$ are piecewise constant, nor why they take distinct values. A justification or reference is needed.
>
> Appendix D contains a full proof of Theorem 3. It explains in detail why $F_{k}$ and $G_{k}$ are piecewise constant. Besides that there is a Monte Carlo experiment in Appendix C that also contains illustrations that imply $F_{k}$, $G_k$ are piecewise constant.
>
> To give a simple illustration, let's consider $D=2, k=2$ and point $a_{1} = 1, a_{2} = 100$. For this point the eigenvalue reads $\lambda_{i_1, i_2} = \left(\pi i_1\right)^2 + 100\left(\pi i_2\right)^2$. Eigenvectors $\phi_{i_1, i_2} = \sin(i_1 \pi x_1) \sin(i_2 \pi x_2)$ are defined by two integers $i_1, i_2 = 1, 2, \dots$. When eigenvectors are ordered by the magnitude of their eigenvalues, for the case considered, we have the following order
> $$
> (1, 1), (2, 1), (3, 1), (4, 1), \dots
> $$
>
> This is because the coefficient before $i_1^2$ is $\pi^2$ is much smaller than $100 \pi^2$.
>
> The order suggest that $F_2(1, 100) = \sin(2\pi x_1)\sin(\pi x_2)$. Now, what happens if we change second coefficient? For example we can consider $a_2 = 100 - R$ with $R\in(-\infty, 100)$? The eigenvalues become $\lambda_{i_1, i_2} = \left(\pi i_1\right)^2 + (100 - R)\left(\pi i_2\right)^2$. Evidently for all negative $R$ the order of first two eigenvectors is not going to change. This implies $F_2(1, 100 - R) = \sin(2\pi x_1)\sin(\pi x_2)$ for all negative $R$. Now, if $R$ is positive the order of two first eigenvectors is going to change only if $1^2 + (100 - R)2^2 < 2^2 + (100 - R) 1^2$, that is $\lambda_{1, 2} < \lambda_{2, 1}$. This happens when $R > 99$. With that we obtain
> $$
> F(1, 100 - R) =
> \begin{cases}
> \sin(2\pi x_1)\sin(\pi x_2), \text{if }R \leq 99;\\
> \sin(\pi x_1)\sin(2\pi x_2), \text{if } 99 < R < 100.
> \end{cases}
> $$
> A more general proof appears in Appendix D.

---

> ### Author Response · Authors · 2025-11-20
> **part 4**
>
> > The description of the SUBSPACE EMBEDDING technique is very unclear. How does it work in practice? Does it imply that the learnable data consist of higher-dimensional subspaces, and if so, how are they embedded and compared?
>
> The definition of subspace embedding technique appears in the first sentence of Section 3.2:
> "In the definition of subspace regression problem (1) we allow to approximate function $\mathbb{R}^{p}\rightarrow \text{Gr}(k, n)$ by function $\mathbb{R}^{p}\rightarrow \text{Gr}(r, n)$ where $r\geq k$. We call this strategy subspace embedding."
>
> It is an unambiguous definition that suggests we approximate a subspace of dimension $k$ by subspace of potentially larger dimension $r$.
>
> To explain how this works in practice, we will consider one dimensional mapping $V:\mathbb{R}\rightarrow \text{Gr}(5, 100)$. We access this mapping by observing $V(x_1) = V_{1},\dots, V(x_N) = V_{N}$, that is $N$  full rank real matrices of size $100\times 5.$ Columns of matrix $V_i$ provide particular basis for the element of $\text{Gr}(5, 100)$ at the point $x_i$. This means we are actually not interested in approximating particular $V_i$ but only linear combinations of its columns. In other words, $V_{1} G_{1},\dots, V_{N}G_{N}$ represents the same data as $V_{1}, \dots, V_{N}$ for arbitrary invertible matrices $G_{1}, \dots, G_N$.
>
> Subspace embedding technique consists of approximating $V(x)$ by mapping $W:\mathbb{R}\rightarrow \text{Gr}(10, 100)$, where $10 > 5$ is chosen for the sake of example. To learn mapping $W$ we choose neural network architecture that takes a real number $x$ as an input and outputs a matrix of size $100\times 10$. We denote such a network $W_{\theta}$, where $\theta$ is a set of parameters.
>
> This network $W_{\theta}$ is trained with one of the loss functions specified in Section 3.1. As an example we will consider computation of $L\_1(A, B) = 5 - \left\|Q\_{B}^\top Q_{A}\right\|\_{F}^2$, where $A = Q\_{A} R\_{A}$ and $B = Q\_{B} R\_{B}$ are reduced QR decompositions of $A \in \mathbb{R}^{100\times 5}$ and $B\in \mathbb{R}^{100 \times 10}$ respectively. As usual we select a minibatch of samples $\left(x_j, V_{j}\right)$, evaluate loss for each sample, average, compute gradient and update network parameters. The only potentially confusing part here is how the loss is computed for matrices of distinct sizes.
>
> Note that the loss is well defined for matrices of distinct sizes. Indeed, for $W_{i} := W_{\theta}(x_i)$ we have $W_{i} = Q_{W_{i}} R_{W_{i}}$ with $Q_{W_i} \in \mathbb{R}^{100\times 10}$, $R_{W_i}\in \mathbb{R}^{10\times 10}$, for $V_i$ we have $V_{i} = Q_{V_{i}} R_{V_{i}}$ with $Q_{V_i} \in \mathbb{R}^{100\times 5}$, $R_{V_i}\in \mathbb{R}^{5\times 5}$. The matrix $Q_{W_{i}}^{\top}Q_{V_{i}}$ has size $\mathbb{R}^{10\times 5}$ and its Frobenius norm is $\sum_{n=1}^{10}\sum_{m=1}^{5} \left(\left(Q_{W_{i}}^{\top}Q_{V_{i}}\right)_{nm}\right)^2.$
>
> Also note that $5 - \left\|Q_{W_{i}}^\top Q_{V_{i}}\right\|_{F}^2 = 0$ if and only if any column of $V_i$ can be represented as a linear combination of columns of $W_i$. In other words, if and only if $\text{range } V_i \subset \text{range }W_i$.
>
> How are the subspaces of different sizes compared? There are many standard options. Two of them explained in Appendix H are:
> 1. Cosines of principal angles between subspaces.
> 2. Relative reconstruction error $\left\|v_i - Q_{W}Q_{W}^{\top} v_i\right\|_2 \big/ \left\|v_i\right\|_2$.
>
> We use none of these metrics in the main text of the article for several reasons: (i) these metrics may look unfamiliar, (ii) it takes time to explain them, (iii) their values are often non-intuitive.
>
> In place of native geometric metrics we use practical ones. In all cases the quality of subspace is accessed by measuring error of performance on the selected task. For example, if we train a neural network to predict subspace that is later used to solve PDE, the quality of subspace is the relative error obtained after we used this subspace to solve PDE. Similarly, if we use subspace to build a reduced model for eigenproblem, the quality of the subspace is accessed by looking on the relative error of eigenvectors computed with the help of this subspace.
>
> > In the tables of numerical results, the meaning of the reported percentages is not explained. Do they represent accuracy, relative error, or improvement over a baseline?
>
> We agree that this part was not explained well in the article. We add a clarification in the beginning of Section 4:
>
> "For most problems we report relative error measured in percentages. For predicted quantity $\widetilde{v}$ and ground truth value $v$ it reads $100\\%\times\left\\|v - \widetilde{v}\right\\|_2 \big/\left\\| v\right\\|_2$. For eigenvalue problems the numerator is $\mathbb{Z}_2$ adjusted as explained below. For iterative methods we report convergence plots for relative error, without additional factor $100\\%$."

---

> > ### Comment · Reviewer_fKUi · 2025-11-26
> >
> > The authors added an experiment showing a clear performance gain. While the presentation could still be improved, the new evidence strengthens the submission, and I am raising my score.

---

### Official Review · Reviewer_LUwp · 2025-11-01

**Soundness:** 4
**Presentation:** 4
**Contribution:** 3
**Rating:** 8
**Confidence:** 4

**Summary:**

The paper introduces Subspace Regression, a learning framework in which the target is a linear subspace rather than a vector or a scalar. It arises naturally in reduced-order modeling (ROM), eigenspace approximation, deflation methods, and parametric PDEs. It formulates the subspace regression on the Grassmann manifold $Gr(k, n)$, proposes two new loss functions invariant under right $GL(k, V)$ actions, and proposes the subspace embedding technique, which predicts a higher-dimensional subspace that contains the target, theoretically shown to smooth the mapping and reduce complexity. I mainly focused on checking the proofs of the theorems and the examples, which look very rigorous to me.

**Strengths:**

1. Formally consider the subspace regression problem with differential geometry.
2. The list of example problems (Sec. 2.2) to solve with subspace regression is very valuable and illustrative, IMHO.
3. I personally like the writing quite a bit, which is concise and right to the point; not much redundancy in the sentences.
4. Theorem 1–3 rigorously justify the loss functions and the subspace embedding principle using differential geometry and asymptotic complexity arguments.
5. Comparisons with classical interpolation, FNO, DeepONet, and POD variants show consistent advantages, especially in high-dimensional parameter spaces.

**Weaknesses:**

1. Tiny suggestion: for QR factorization, it is more precise to say $A = Q_A R_A$, where $Q_A\in \mathbb{R}^{n\times k}$ since for a tall matrix $A$, you can factorize it into a unitary $Q \in \mathbb{R}^{n\times n}$.
2. Another small suggestion, in Thm. 1, for $L_1$, if $A = B \cdot G$, for some $G\in GL(k, V)$, then $L_1(A, B) = 0$, which explains the $p$ term in the loss, which might look a bit weird at first look.
3. Yet another thing: for the parametrization of $V$, you wrote $V(r) \colon \mathbb{R}^p \rightarrow Gr(k, n)$ (which I understand), which, however, confused me at the first glance: I thought it mean for each $r$, $V(r)$ is a map from $\mathbb{R}^p$ to $Gr(k, n)$. Better to write $V \colon \mathbb{R}^p \rightarrow Gr(k, n)$.
4. Thm. 2 shows that larger subspaces help with reducing the magnitude of the tangent vectors, but it doesn’t quantify why it helps with optimizing the loss function, e.g., I think Lipschitz analysis of the tangent vectors might help.
5. Many results are reported as single-run metrics without confidence intervals or error bars.

**Questions:**

1. Just a tiny suggestion: Sec. 2.1, I suggest using [W] to denote the equivalence class.
2. Line 88, why the parameteric model maps to Gr(r, n), r \geq k, which is more expressive than needed?
3. In the Schrödinger eigenproblem, I do not really get why it is a subspace regression problem. Do you wish to learn the coefficients or the basis $\phi_i$? Also, in line 113, what is $f$? $\alpha_i$ should be complex-valued.
4. How is the embedding dimension $r > k$ chosen in practice?
5.  How sensitive is the training to the choice of r.v. in the stochastic loss $L_2$?
6. For high-dimensional PDEs, does the cost of orthogonalization or QR decomposition outweigh the benefits of manifold losses?

---

> ### Author Response · Authors · 2025-11-20
> **part 1**
>
> We would like to thank the reviewer for many useful suggestions. In response we introduce several changes detailed below.
>
> > Tiny suggestion: for QR factorization, it is more precise to say $A = Q_{A} R_{A}$, where $Q_{A}\in\mathbb{R}^{n\times k}$ since for a tall matrix $A$, you can factorize it into a unitary $Q \in \mathbb{R}^{n\times n}$.
>
> First time QR decomposition appears in Theorem 1. We agree that it is important that reduced form of QR should be used. We add that a reduced QR decomposition is assumed and provide its definition in the footnote.
>
> > Another small suggestion, in Thm. 1, for $L_1$, if $A = B \cdot G$, for some $G\in GL(k, V)$, then $L_1(A, B) = 0$, which explains the $p$ term in the loss, which might look a bit weird at first look.
>
> We agree that $p$ can lead to some confusion, so we introduce clarification why $p$ is needed right after Theorem 1.
>
> > Yet another thing: for the parametrization of $V$, you wrote $V(r) \colon \mathbb{R}^p \rightarrow Gr(k, n)$ (which I understand), which, however, confused me at the first glance: I thought it mean for each $r$, $V(r)$ is a map from $\mathbb{R}^p$ to $Gr(k, n)$. Better to write $V \colon \mathbb{R}^p \rightarrow Gr(k, n)$.
>
> This notation appears on line 85 and in Theorem 2. Following the reviewer suggestion we now write $V:\mathbb{R}\rightarrow \text{Gr}(k, n)$ and $W:\mathbb{R}\rightarrow \text{Gr}(r, n)$.

---

> ### Author Response · Authors · 2025-11-20
> **part 2**
>
> > Thm. 2 shows that larger subspaces help with reducing the magnitude of the tangent vectors, but it doesn’t quantify why it helps with optimizing the loss function, e.g., I think Lipschitz analysis of the tangent vectors might help.
>
> We agree that Theorem 2 does not imply that the loss function is easier to optimize. In our view Theorem 2 suggests that embedding technique can lead to smoother function. The smoothness is important for two reasons: (i) in general smoother functions are easier to approximate, with constant function being an extreme example, (ii) there is a general observation -- called f-principle -- that neural networks first learn low frequencies.
>
> Theorem 2, however, is a constructive result that shows improvement in smoothness for the embedding of geodesics. It provides only circumstantial evidence that the same may happen when neural networks are trained. To gather direct evidence we perform several experiments and report results in Appendix F.4. The results are briefly introduced in Section 4.1 in the main text. Here we provide an excerpt of these results.
>
> All indicators depend on two features: $f_1$, $f_2$ and real number $l\in[0, 1]$. We compute indicators for trained networks (loss function $L_2(A, B; z)$) and several $l$. Each indicator is evaluated for $1000$ randomly selected pairs of features $f_1, f_2$ and average is reported.
>
> Taylor indicator $T[f_1, f_2; l] = \left\|\mathcal{N}(f_1 + l f_2) -  \mathcal{N}(f_1) - \left.\frac{d}{dl} \mathcal{N}(f_1 + l f_2)\right|_{l=0}l\right\|_2 \big/ \left\|\mathcal{N}(f_1 + l f_2)\right\|_2$.
> Taylor indicator is a relative error of linear model. One expects that: (i) unless $\mathcal{N}$ is a linear function, when $l$ increases relative error also increases; (ii) for smoother maps Taylor indicator should be smaller.
>
> Results for $k_1 = k_2$, $D=2$, elliptic eigenproblem.
>
> | $N_{\text{subspace}}$ | $l=0.25$ | $l=0.5$ | $l=0.75$ | $l=1.0$ |
> | --------------------- | -------- | ------- | -------- | ------- |
> | 10                    | 2.41     | 5.1     | 7.84     | 10.71   |
> | 20                    | 2.12     | 4.57    | 7.05     | 9.68    |
> | 30                    | 1.88     | 4.09    | 6.34     | 8.73    |
> | 40                    | 1.76     | 3.84    | 5.98     | 8.22    |
>
> Results for $k_1 \neq k_2$, $D=2$, elliptic eigenproblem.
>
> | $N_{\text{subspace}}$ | $l=0.25$ | $l=0.5$ | $l=0.75$ | $l=1.0$ |
> | --------------------- | -------- | ------- | -------- | ------- |
> | 10                    | 1.8      | 3.87    | 5.99     | 8.21    |
> | 20                    | 1.72     | 3.77    | 5.86     | 8.08    |
> | 30                    | 1.71     | 3.77    | 5.87     | 8.08    |
> | 40                    | 1.42     | 3.15    | 4.94     | 6.8     |
>
> In both cases we observe that smoothness improves with the increase of subspace size.
>
> Mean cosine indicator $C[f_1, f_2;l] = \frac{1}{D}\sum_{i=1}^{D} \cos_{i}(\mathcal{N}(f_1 + l f_2), t_1)$, where $\cos_i(A, B)$ are cosines of principle angles computed with SVD and $t_1$ is target at point $f_1$. Mean cosine indicator is a mean value of cosines between target and network prediction computes in nearby points $f_1 +l f_2$ and $f_1$. If smoothness of the learned map improves, we expect $C[f_1, f_2; l]$ to increase toward $1$.
>
> Results for $k_1 = k_2$, $D=2$, elliptic eigenproblem.
>
> | $N_{\text{subspace}}$ | $l=0.25$ | $l=0.5$ | $l=0.75$ | $l=1.0$ |
> | --------------------- | -------- | ------- | -------- | ------- |
> | 10                    | 0.81     | 0.74    | 0.67     | 0.51    |
> | 20                    | 0.94     | 0.88    | 0.82     | 0.68    |
> | 30                    | 0.97     | 0.93    | 0.88     | 0.78    |
> | 40                    | 0.98     | 0.95    | 0.91     | 0.82    |
>
>
> Results for $k_1 \neq k_2$, $D=2$, elliptic eigenproblem.
>
> | $N_{\text{subspace}}$ | $l=0.25$ | $l=0.5$ | $l=0.75$ | $l=1.0$ |
> | --------------------- | -------- | ------- | -------- | ------- |
> | 10                    | 0.82     | 0.74    | 0.67     | 0.53    |
> | 20                    | 0.94     | 0.88    | 0.83     | 0.72    |
> | 30                    | 0.97     | 0.93    | 0.89     | 0.81    |
> | 40                    | 0.99     | 0.96    | 0.93     | 0.87    |
>
> Results clearly indicate that the learned map $\mathcal{N}(f) \equiv W(f)$ becomes more uniform, averaging information about subspaces $V(f_1)$ and $V(f_2)$ for distinct features. It is instructive to compare results $l=1.0$ with mean cosines $\frac{1}{D}\sum_{i=1}^{D} \cos_{i}(t_1, t_2)$ computed for $1000$ randomly selected targets
>
> | dataset        | mean cos $\pm$ std |
> | -------------- | ------------------ |
> | $k_1 = k_2$    | $0.51\pm0.18$      |
> | $k_1 \neq k_2$ | $0.53\pm0.12$      |

---

> ### Author Response · Authors · 2025-11-20
> **part 3**
>
> > Many results are reported as single-run metrics without confidence intervals or error bars.
>
> We acknowledge that it is not ideal to report results without error bars. To study sensitivity to train-test split we performed several additional experiments. Full results are available in Appendix J in the revised version of the manuscript. Brief discussion is introduced in the beginning of Section 4. We find that: (i) sensitivity to train-test split is not pronounced enough to change main conclusions; (ii) sensitivity to hyperparameter search is much higher.
>
> As an illustration we report several results here.
>
> Loss $L(A, B; z)$ for elliptic equation with $k_1 = k_2$
>
> | $N_{\text{subspace}}$ | train error $\pm$ std, $\\%$ | test error $\pm$ std. $\\%$ |
> | --------------------- | -------------------------- | ------------------------- |
> | $10$                  | $24.37\pm1.47$             | $30.38\pm1.34$            |
> | $20$                  | $5.69\pm0.53$              | $6.46\pm0.26$             |
> | $30$                  | $3.34\pm0.26$              | $3.79\pm0.35$             |
> | $40$                  | $1.84\pm0.12$              | $2.12\pm0.11$             |
>
> Loss $L(A, B; z)$ for Burgers equation
>
> | $N_{\text{subspace}}$ | train error $\pm$ std, $\\%$ | test error $\pm$ std, $\\%$ |
> | --------------------- | -------------------------- | ------------------------- |
> | $10$                  | $25.5\pm1.17$              | $25.71\pm1.72$            |
> | $20$                  | $11.15\pm1.2$              | $11.0\pm1.43$             |
> | $30$                  | $7.04\pm0.35$              | $7.01\pm0.64$             |
> | $40$                  | $4.56\pm0.32$              | $4.72\pm0.61$             |
>
> For comparison we also provide sensitivity for sweep over hyperparameters
>
> Loss $L(A, B; z)$ for elliptic equation with $k_1 = k_2$
>
> | $N_{\text{subspace}}$ | train error $[\min, \max]$, $\\%$ | test error $[\min, \max]$, $\\%$ |
> | --------------------- | -------------------------------- | ------------------------------- |
> | $10$                  | $[23.81, 37.8]$                  | $[28.67, 39.05]$                |
> | $20$                  | $[4.74, 100.0]$                  | $[5.51, 100.0]$                 |
> | $30$                  | $[2.77, 8.6]$                    | $[3.33, 9.06]$                  |
> | $40$                  | $[2.06, 7.05]$                   | $[2.31, 7.34]$                  |
>
> Note that for elliptic equations the search was performed on the downsampled datasets on grid $32\times32$ with $500$ epoch. In contrast, results for train-test split sensitivity are obtained on grid $100\times 100$ with $1000$ epoch. This explains the seeming inconsistency between best results reported for these two cases.
>
> Loss $L(A, B; z)$ for Burgers equation
>
> | $N_{\text{subspace}}$ | train error $[\min, \max]$, $\\%$ | test error $[\min, \max]$, $\\%$ |
> | --------------------- | -------------------------------- | ------------------------------- |
> | $10$                  | $[22.01, 32.64]$                 | $[23.79, 34.62]$                |
> | $20$                  | $[9.53, 20.14]$                  | $[10.06, 21.05]$                |
> | $30$                  | $[5.58, 21.81]$                  | $[5.84, 21.7]$                  |
> | $40$                  | $[4.3, 47.43]$                   | $[4.31, 283.33]$                |
>
> For both Burgers and elliptic equations we find sensitivity to train-test split to be small. In contrast, sensitivity to the choice of hyperparameters is high.

---

> ### Author Response · Authors · 2025-11-20
> **part 4**
>
> > Just a tiny suggestion: Sec. 2.1, I suggest using [W] to denote the equivalence class.
>
> The notation for equivalence class used in the article is borrowed from Gabriele Ciaramella, Martin J Gander, and Tommaso Vanzan. A gentle introduction to interpolation on the Grassmann manifold 2025. We are not particularly attached to this notation, but we can not use $[W]$ for equivalence class, because square brackets are reserved for averages $\mathbb{E}[\cdot]$ and some readers may be confused about the meaning of equation (1) and average in Theorem 1.
>
> > Line 88, why the parameteric model maps to $Gr(r, n), r \geq k$, which is more expressive than needed?
>
> We allow for mapping to the subspace of larger size to accommodate the subspace embedding technique introduced later. To clarify this part we include an additional footnote.
>
> >In the Schrödinger eigenproblem, I do not really get why it is a subspace regression problem. Do you wish to learn the coefficients or the basis $\phi_i$? Also, in line 113, what is $f$? $\alpha_i$ should be complex-valued.
>
> We agree that this part can be slightly confusing.
>
> In principle prediction of eigenvectors can be done directly. This is explained in Section 4.1. However, results reported in Figure 1 also suggest that this problem is too complex already for the third eigenvector.
>
> We add clarification that a  natural alternative is to predict a subspace spanned by first $K$ eigenvectors and later use this subspace to solve the reduced problem. We also change the notation for set defined formerly on line 113 to $U(x) \rightarrow \left\\{f(x): f(x)=\sum\_{i=1}^{K}\alpha_i\phi_i(x), \alpha_i \in \mathbb{C}\right\\}$. So it is a set of functions that can be represented as linear combinations of first $K$ eigenvectors. They form a subspace of dimension $K$.
>
> > How is the embedding dimension $r > k$ chosen in practice?
>
> According to our results the larger $r$ the better, but we expect that the improvement will saturate since current architecture rarely reaches relative error below $10^{-3}$ on operator learning problems. The other consideration besides accuracy is cost. Clearly, with the growth of $r$ the complexity of reduced problems increases.
>
>
> > How sensitive is the training to the choice of r.v. in the stochastic loss $L_2$?
>
> We tried and found no significant improvement when the number of vectors increases. On the contrary we found that lower numbers of vectors often provide more regularisation. The effect is not pronounced enough, so we do not include the results with different numbers of random vectors.
>
> > For high-dimensional PDEs, does the cost of orthogonalization or QR decomposition outweigh the benefits of manifold losses?
>
> For high dimensional PDEs the problem becomes extremely complicated. Curse of dimensionality (COD) appears twice: (i) space of parameters is high-dimensional, (ii) number of dimensions is large, so it becomes harder to represent functions. For the former reason QR decomposition will certainly be problematic. Besides that it is not clear whether neural networks overcome the curse of dimensionality. It is known that the curse is absent for parametric PDEs (the first appearance of COD) [1], [2], but we are not aware whether this is the case when in addition the number of spatial dimensions is large.
>
> [1] Exponential Convergence of Deep Operator Networks, https://arxiv.org/abs/2112.08125
>
> [2] Operator learning with PCA-Net: upper and lower complexity bounds, https://arxiv.org/abs/2303.16317

---

### Author Response · Authors · 2025-11-20

We would like to thank all reviewers for reading the manuscript carefully and providing the feedback. We incorporated many changes which we detailed in the individual responses. Here we provide a general summary of the changes in the revised manuscript.

Changelog:
1. We corrected several grammatical inaccuracies in the appendices. Since the appendix is optional we do not highlight changes made there.
2. We also introduced many small changes and added numerous clarifications in the main text. These changes are marked in blue in the revision.
3. To access sensitivity to train-test split we perform additional experiment and summarised it in Appendix J
4. We confirm that neural networks trained on resolution $n$ generalise to higher resolutions $m>n$. The experiment is explained in Appendix F.4
5. We confirm numerically that subspace embedding leads to smoother function. This is explained in Appendix F.5
6. We show that subspace regression can speed up and stabilize convergence of LOBPCG. Results appear in Appendix F.6

---

### Comment · Area_Chair_EDsS · 2025-11-26
**reminder**

Dear Reviewers,

The authors have now posted their responses to your comments. As the next step in the discussion phase, please take a moment to review their rebuttal and engage with them through the discussion forum. Thank you for your continued effort and thoughtful contributions to this review.

Best,

Your AC

---

### Author Response · Authors · 2025-12-02
**discussion summary**

For the convenience of AC we briefly summarise the discussion with reviewers and changes introduced to the revised manuscript.

**Reviewer LUwp** raised two main concerns:
1. Theorem 2 explains why accuracy improves only indirectly.

   We address this by performing additional experiments reported in Appendix F.4, summarised in [this comment](https://openreview.net/forum?id=HF60Lu1Maj&noteId=V5eUvGxm9P)
2. Reported results do not have error bars.

   We perform a series of additional experiments to study sensitivity to train-test split. Results are reported in Appendix J and in [this comment](https://openreview.net/forum?id=HF60Lu1Maj&noteId=ujlBRPwgaZ).

The reviewer did not appear before the rebuttal discussion was stopped after the incident.

---

**Reviewer fKUi** raised two concerns:
1. Several parts were not clearly explained.

   We supply additional explanations in many parts of the text. They are highlighted in blue colour in the revised manuscript.
2. Additional comparison with classical techniques is needed for eigenproblems.

   We compared LOBPCG with LOBPCG initialised with subspace regression. Appendix F.5. and Section 4.1. of revised manuscript contains the description and the results of the experiment. They are also summarised in [this comment](https://openreview.net/forum?id=HF60Lu1Maj&noteId=gJkg8bG40r).

The reviewer acknowledged that we addressed main concerns in [the reply](https://openreview.net/forum?id=HF60Lu1Maj&noteId=GfCivgJ2DZ). As a result the score was increased from 4 to 6.

---

**Reviewer MfjB** noted a few typos and unclear parts; asked for several clarifications, which we provided in a series of replies. Notably, we perform additional experiments on “discretisation invariance”. It is summarised in [this message](https://openreview.net/forum?id=HF60Lu1Maj&noteId=pxTEYomXYg) and in Appendix F.4 of the revised manuscript.

In response to our rebuttal the reviewer increased the score from 8 to 10 as indicated in [this reply](https://openreview.net/forum?id=HF60Lu1Maj&noteId=9uLMQKZtr6).

---

**Reviewer jxox** raised several minor questions and asked for a dedicated related research review. We supplied such a review in [the reply](https://openreview.net/forum?id=HF60Lu1Maj&noteId=Fw4Fo1YJ8f), but did not incorporate it into the text of the revised manuscript.

The reviewer did not appear before the discussion stops.

---

### Meta-Review · Area_Chair_Wav9 · 2025-12-28

**Summary:**

This submission introduces a subspace regression framework where the prediction target is a linear subspace on the Grassmann manifold, motivated by reduced-order modeling, eigenspace approximation, and related operator-learning settings. The paper makes a clear methodological contribution via manifold-aware loss functions and the “subspace embedding” idea (predicting a higher-dimensional subspace that contains the target), supported by theoretical arguments about smoothness/complexity and by broad empirical evaluations. The reviews collectively view the technical core as promising and, in parts, rigorous, while the main reservations concerned (i) whether the theoretical discussion of why embedding helps translates into optimization and learning benefits, (ii) insufficient uncertainty quantification in reported metrics, (iii) clarity/notation and missing definitions, and (iv) the need for stronger classical non-learning baselines in eigenproblems. The rebuttal and revision address the most consequential issues with additional experiments, clearer exposition, and added baselines, leading to an overall positive assessment suitable for acceptance as a poster.

**Reviewer Concerns:**

A key concern was that Theorem 2 provides only indirect justification for improved performance; the authors explicitly acknowledged this limitation and added targeted experiments intended to provide more direct evidence that embedding yields smoother learned maps, strengthening the empirical rationale. Another major concern was the lack of error bars and reliance on single-run metrics; the authors added train–test split sensitivity studies and reported variability, which improves credibility of the empirical claims (while also noting that hyperparameter sensitivity can dominate). Multiple presentation issues were raised (notation collisions, missing definitions, unclear descriptions of subspace embedding, and ambiguity in what the reported “%” metrics represent); the revision appears to address these with renaming, added definitions (including “relative error”), and expanded explanations of how embedding is implemented and evaluated. The “no classical baselines” critique for eigenproblems was directly addressed by adding experiments with a strong non-learning method (LOBPCG) and showing that subspace regression can provide useful warm-starts that improve convergence and accuracy. Remaining weaknesses are primarily editorial and scope-related: some explanations may still be dense for non-specialists, the broader question of computational overhead (e.g., orthogonalization costs in very high-dimensional PDE settings) is discussed but not fully resolved with comprehensive profiling, and at least one reviewer-requested related-work discussion was provided in replies but not fully integrated into the main text.

**Reviewer Scores:**

Reviewer LUwp was already strongly positive (accept-level) and, given the nature of their comments and the authors’ clarifications, I would expect their score to remain essentially unchanged. Reviewer fKUi initially expressed below-threshold concerns focused on clarity and missing comparisons; after the added baseline evidence and the expanded explanations/definitions, I would expect a material upward adjustment into a weak-accept range if they had participated fully in the discussion. Reviewer MfjB was positive and explicitly indicated willingness to raise the score contingent on clarifications; given that the authors added the requested definition of relative error and corrected missing/unclear elements, I would expect an upward score revision reflecting stronger confidence in the presentation and empirical reporting. Reviewer jxox raised mostly minor questions and requested a dedicated related-work review; because their follow-up participation appears limited, I would conservatively assume their score would stay near the original level, with a plausible small increase if the related-work discussion were incorporated into the manuscript rather than confined to the forum.

---

### Decision · Program_Chairs · 2026-01-26

Accept (Poster)